# SEMANTIX: AN ENERGY-GUIDED SAMPLER FOR SEMANTIC STYLE TRANSFER

**Huiang He** *
South China University of Technology
mshuianghe@mail.scut.edu.cn

**Minghui Hu** *
SpellBrush & Nanyang Technological University
e200008@e.ntu.edu.sg

**Chuanxia Zheng**
VGG, University of Oxford
cxzheng@robots.ox.ac.uk

**Chaoyue Wang**
The University of Sydney
chaoyue.wang@outlook.com

**Tat-Jen Cham**
College of Computing and Data Science
Nanyang Technological University
ASTJCham@ntu.edu.sg

## ABSTRACT

Recent advances in style and appearance transfer are impressive, but most methods isolate global style and local appearance transfer, neglecting semantic correspondence. Additionally, image and video tasks are typically handled in isolation, with little focus on integrating them for video transfer. To address these limitations, we introduce a novel task, *Semantic Style Transfer*, which involves transferring style and appearance features from a reference image to a target visual content based on semantic correspondence. We subsequently propose a training-free method, *Semantix*, an energy-guided sampler designed for Semantic Style Transfer that simultaneously guides both style and appearance transfer based on semantic understanding capacity of pre-trained diffusion models. Additionally, as a sampler, *Semantix* can be seamlessly applied to both image and video models, enabling semantic style transfer to be generic across various visual media. Specifically, once inverting both reference and context images or videos to noise space by SDEs, *Semantix* utilizes a meticulously crafted energy function to guide the sampling process, including three key components: *Style Feature Guidance*, *Spatial Feature Guidance* and *Semantic Distance* as a regularisation term. Experimental results demonstrate that *Semantix* not only effectively accomplishes the task of semantic style transfer across images and videos, but also surpasses existing state-of-the-art solutions in both fields.

## 1 INTRODUCTION

The vision community has rapidly improved the quality of image and video generation over a short period. In particular, some powerful baseline systems for Text-to-Image (Podell et al., 2023; Saharia et al., 2022; Nichol et al., 2021; Rombach et al., 2022; Ramesh et al., 2022) and Text-to-Video (Blattmann et al., 2023; Guo et al., 2023; Girdhar et al., 2023) have been proposed, which contributes a series of applications such as controllable generation in ControlNet (Zhang et al., 2023a; Hu et al., 2023), IPAdapter (Ye et al., 2023) and InstantID (Wang et al., 2024c). Among these innovations, one significant field is visual transfer, which modifies a context image to fit the style or appearance of the reference image, while preserving the original content or structure.

Prior works have extensively explored visual transfer. However, they typically focus on two distinct scenarios: (1) Style transfer (Gatys et al., 2015; 2016; Huang and Belongie, 2017; Li et al., 2017; Liu et al., 2021; Deng et al., 2022; Wang et al., 2023b; 2024a; Ye et al., 2023) that utilizes the global stylistic features for the entire image style modification, but overly emphasize the overarching style of the reference images; and (2) Appearance transfer (Isola et al., 2017; Zhu et al., 2017; Park et al., 2020a;b; Zheng et al., 2021; Mou et al., 2023; Alaluf et al., 2023; Wang et al., 2024b) that conveys

---

*equal contribution

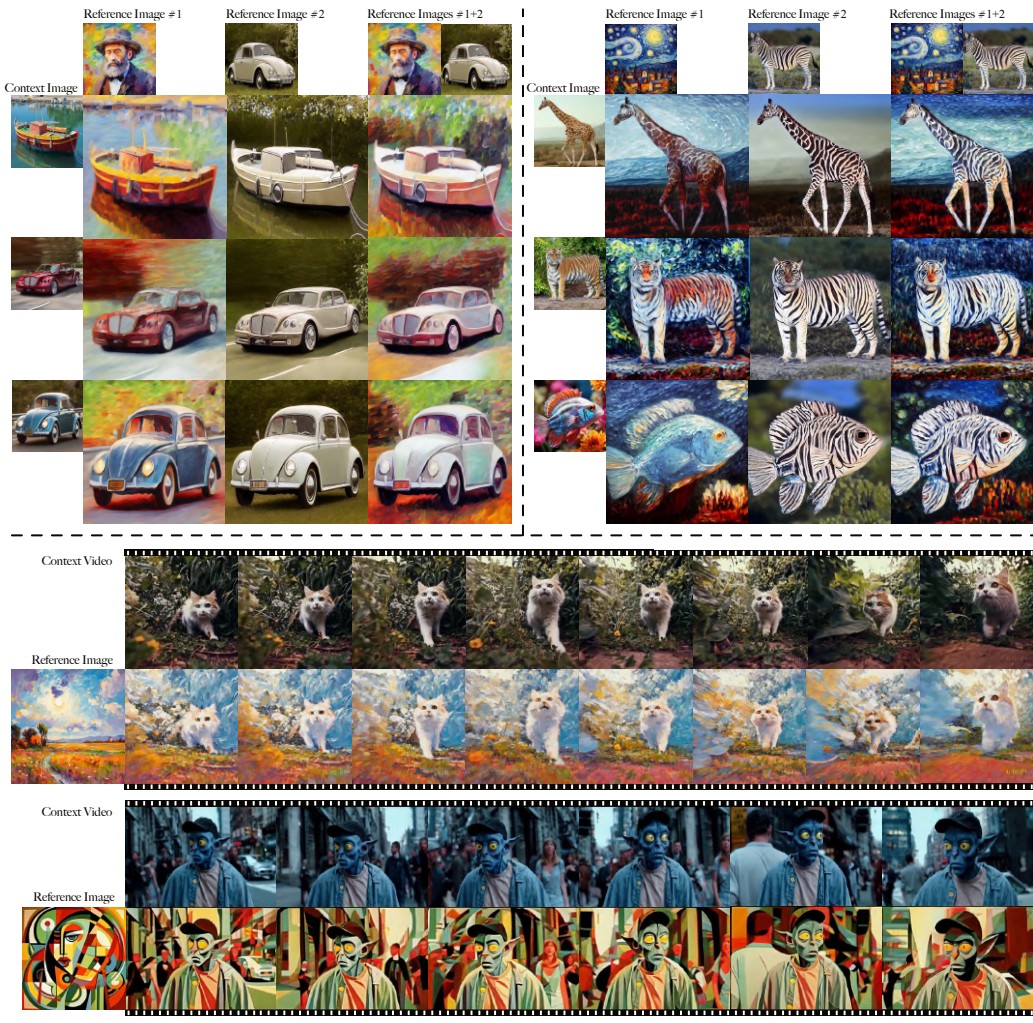

Figure 1: **Examples of our *Semantix*.** Given a visual context and a reference image (Top examples), *Semantix* can perform *Semantic Style Transfer* based on the semantic correspondence. Besides, our Semantix also can be directly adapted for the videos (Bottom examples) without the need of additional modification. It is important to emphasize that, as a sampler, Semantix directly leverages *the knowledge from the pretrained model* to guide the sampling process based on our proposed energy function for Semantic Style Transfer, *without the need for any additional training or optimization.*

the object appearance from the reference to the context image, but exhibits only limited sensitivity to the overall perceptual style. Besides, both of them ignore semantic alignment and video continuity during the transfer process. These factors destroy the video continuity and lead to global style transfer risking content leakage, while local appearance transfer may disrupt structural integrity.

We observe that the tasks of style transfer and appearance transfer share similarities in their underlying objectives: to transfer relevant information from the reference visual content to the context visual content. We assume that a feature transfer task guided by semantic alignment can better integrate these two tasks, mitigating the risks of content leakage and structural disruption. Thus we define a task termed *Semantic Style Transfer* as: given a context visual content, *e.g.*, image or video, and a reference image with style and (or) appearance features, the objective of Semantic Style Transfer is to *analyse and transfer* the features from the reference image to the context visual content *through precise semantic mapping*. Specifically, semantic style transfer considers semantic correspondence between context and reference images. When there is a clear semantic correspondence between the context image and the reference image, the transfer is executed based on the semantic correspondence. For instance, as shown in Fig. 1, the body of the giraffe and the zebra exhibit semantic correspondence,

then the visual features of the body of zebra will be applied to the body of giraffe. Conversely, when semantic correspondence is weak, as between village in Van Gogh's style and the giraffe, the style is injected based on the other correlations, *e.g.*, color information or positional information.

To achieve this goal, we propose *Semantix*, an energy-guided sampler, which leverages the strong semantic alignment capabilities of pre-trained diffusion models (Epstein et al., 2023) to transfer features from the reference image to the context visual based on the semantic correspondence without any training or optimization. Initially, we employ SDE Inversion (Huberman-Spiegelglas et al., 2023; Nie et al., 2023) to invert given content into the noise manifold, establishing a conducive foundation for semantic style manipulation. We then introduce a specialized energy function for semantic style transfer that guides the sampling process rather than modify the model structure (Alaluf et al., 2023), thus maintaining the original capabilities of the visual models and supporting video continuity. Our proposed energy function comprises three terms: *i) Style Feature Guidance*, to align the style features with the reference image; *ii) Spatial Feature Guidance*, to maintain spatial coherence with context; and *iii) Semantic Distance*, to regularise the whole function. In particular, within the term of style feature guidance, we initially leverage semantic correspondence in the diffusion model (Tang et al., 2023) and position encoding (Vaswani et al., 2017; Dosovitskiy et al., 2020) to align the output and style features. In the spatial feature guidance component, we directly consider the feature distances at corresponding positions to ensure consistency between the generated content and the context. Lastly, we utilize the distance between the cross-attention maps of the generated content and the given context as a regularisation term to enhance stability.

Integrating these capabilities, *Semantix* effectively transfers the features with semantic from the reference image to the context with precise spatial alignment, satisfying the requirements of Semantic Style Transfer. Additionally, as *Semantix* functions as a sampler for diffusion models, it can be seamlessly applied to various image or video base models. Since these pre-trained models already encapsulate sufficient semantic information and maintain action coherence, Semantix enables training-free semantic style transfer across image and video simply through guided sampling.

In summary, our contributions include the following key points:

- We introduce Semantix, an energy-guided sampler specifically designed for training-free semantic style transfer. We further extend Semantix across images and videos, demonstrating the versatility of energy-guided samplers.

- Experimental results demonstrate that Semantix yields superior results in training-free semantic style transfer across images and videos, surpassing existing solutions in both style and appearance transfer in terms of accuracy and adaptability.

## 2 RELATED WORKS

**Style and Appearance Transfer**   Style transfer infuses the style information of a reference image into a context image to synthesize a stylized context image. Previous convolution-based methods (Huang and Belongie, 2017; Gatys et al., 2016; Johnson et al., 2016; Li et al., 2017; 2018; Park and Lee, 2019; Lai et al., 2017; Gu et al., 2018) and Transformer-based methods (Deng et al., 2022; Wu et al., 2021; Wang et al., 2022; Liu et al., 2021) have successfully facilitated style transfer through the fusion of style and context information. Recently, diffusion-based style transfer has seen significant attention and progress. Some works (Ye et al., 2023; Wang et al., 2024a; 2023b; Sohn et al., 2023) use additional trained networks to extract style features for guiding image synthesis. These methods inject global style features into the context image, altering its overall color and brush strokes but ignoring the necessary semantic correspondence for meaningful style transfer. Other training-based methods (Ruiz et al., 2023a;b; Shi et al., 2023; Wei et al., 2023; Li et al., 2024) require fine-tuning the diffusion model to learn specific styles or introducing new text embeddings for style representation through textual inversion (Gal et al., 2022; Zhang et al., 2023b). These methods are time-consuming and often struggle to balance style injection with context preservation. Some studies (Qi et al., 2024; Wang et al., 2023a; Jeong et al., 2023; Frenkel et al., 2024; Gandikota et al., 2023) attempt to achieve style transfer through decoupling, but they either face challenges in acquiring paired style datasets or risk losing image context. Contrastive learning (Yang et al., 2023; Chen et al., 2021; Zhang et al., 2022; Park et al., 2020a) has also been employed, requiring carefully designed loss functions for optimization. Howerver, all these methods need additional training. Some

training-free methods (Chung et al., 2023; Deng et al., 2023; Hertz et al., 2023; Jeong et al., 2024) induce style transfer by manipulating features within the attention blocks of the diffusion model.

Meanwhile, appearance transfer aims to map appearance from one image to another. Early methods based on GANs (Isola et al., 2017; Zhu et al., 2017; Yi et al., 2017) have faced practical limitations. Other approaches using VAEs (Park et al., 2020b; Liu et al., 2017; Jha et al., 2018; Pidhorskyi et al., 2020) encode images into separate structure and appearance representations to combine features from different images. Recent diffusion-based techniques have significantly advanced appearance transfer (Mou et al., 2023; Epstein et al., 2023; Kwon and Ye, 2022; Alaluf et al., 2023; Wang et al., 2024b). However, these approaches utilize local appearance features for guidance, overlooking precise semantic correspondence between images, leading to inaccurate appearance transfer.

While related dense style transfer work (Ozaydin et al.) introduces the concept of semantics, our definitions and tasks differ significantly, particularly in how semantic features are extracted and used. Specifically, unlike some semantic style transfer methods relying on segmentation labels (Shen et al., 2019; Bhattacharjee et al., 2020) or CLIP visual features (Ozaydin et al.), they lose natural language alignment. Furthermore, such existing works only extract dense semantic features and reuse it in given image. Instead, our work defines a more expansive zero-shot semantic style transfer task focused on generation. In detail, rather than merely reusing the low-level features from a given image, *e.g.*, color, we leveraged a powerful pre-trained diffusion model to generate novel style images that adhere to semantic constraints derived from the features of the given image.

**Semantic Correspondence between Images**    Establishing semantic correspondence between images is crucial. Past methods (Zhang et al., 2021; Zhao et al., 2021) using supervised learning require extensive annotations. To address data limitations, some works (Wang et al., 2020; Rocco et al., 2018; Lee et al., 2019; Seo et al., 2018) use weakly supervised approaches. Recently, self-supervised learning has gained attention, especially with diffusion models. Studies (Tang et al., 2023; Zhang et al., 2024; Caron et al., 2021) leverage these models representation abilities to calculate feature similarities and establish semantic correspondence during the denoising process.

**Energy Functions**    Previous research interprets diffusion models as energy-based models (Liu et al., 2022), where the energy function guides the generation process for precise outputs. The energy function has various applications, such as energy-guided image editing (Mou et al., 2023; 2024; Epstein et al., 2023) and translation (Zhao et al., 2022). The energy function also shows potential in controllable generation, guiding generation through conditions such as sketch (Voynov et al., 2023a), mask (Singh et al., 2023), layout (Chen et al., 2024), concept (Liu et al., 2022) and universal guidance (Bansal et al., 2023; Yu et al., 2023), enabling precise control over the output.

## 3 PRELIMINARIES

**Energy Function**    Diffusion models can be viewed as score-based generative models (Song et al., 2020b). In classifier guidance (Dhariwal and Nichol, 2021; Ho and Salimans, 2022), the gradient of the classifier $\nabla_{x_t} \log p_\phi(y|x_t)$ is used to influence generation. From the perspective of score functions, the condition $y$ can be integrated within a conditional probability $q(x_t|y)$ via an auxiliary score function and expressed as such:

$$\nabla_{x_t} \log q(x_t|y) = \nabla_{x_t} \log \left( \frac{q(y|x_t)q(x_t)}{q(y)} \right) \propto \nabla_{x_t} \log q(x_t) + \nabla_{x_t} \log q(y|x_t), \quad (1)$$

where the first term is viewed as the unconditional denoiser $\epsilon_\theta(x_t; t, \emptyset)$, and the second term can be interpreted as the gradient of the energy function: $\mathcal{E}(x_t; t, y) = \log q(y|x_t)$. Alternatively, classifier-free guidance (CFG) (Ho and Salimans, 2022) can also be used, expressed as:

$$\hat{\epsilon}_t = (1 + \omega)\epsilon_\theta(x_t; t, y) - \omega\epsilon_\theta(x_t; t, \emptyset) \quad (2)$$

where $\omega$ is the classifier-free guidance strength. In fact, the diffusion model can also be interpreted as an energy-based model (Liu et al., 2022), guided by any energy function. One can design an energy function beyond class-based conditioning and use it for guidance (Zhao et al., 2022; Epstein et al., 2023; Bansal et al., 2023; Chen et al., 2024; Yu et al., 2023; Voynov et al., 2023a; Kwon and Ye, 2022). Such guidance provides directional information to guide the diffusion process, and if appropriately

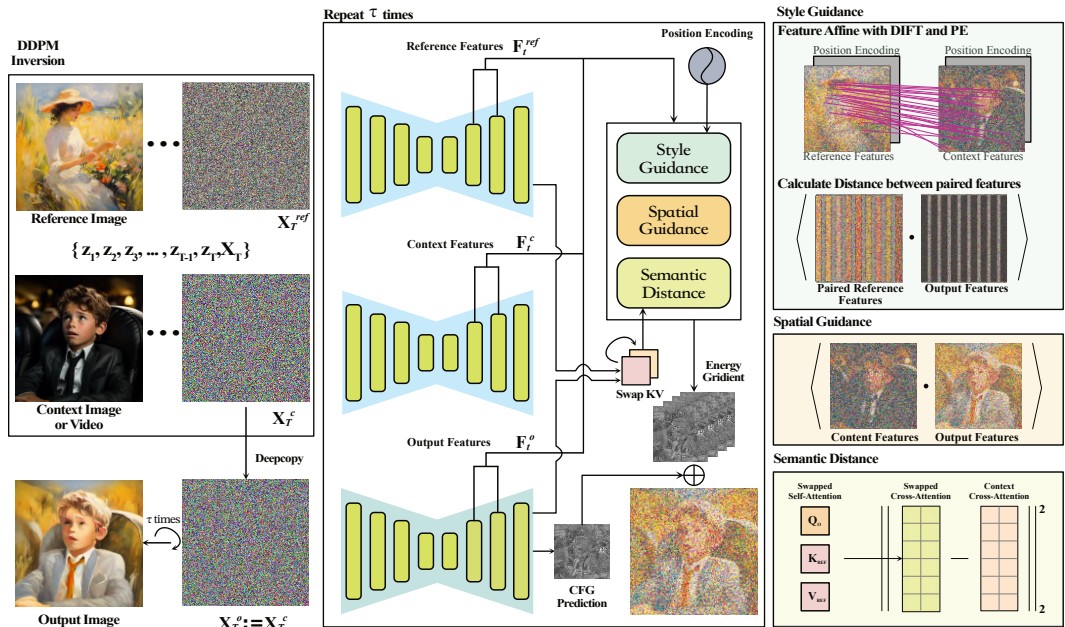

Figure 2: **Overview of Semantix**. Given a reference image $I^{ref}$ and a context image $I^c$ or video $V^c$, we first invert them to the latent $x_T$ through an edit-friendly DDPM inversion. In the denoising process, we then modify the $x_t^{out}$ through the designed energy gradient in every sampling step.

designed, as we shall show in this paper, it can be used to preserve semantic structure while adjusting the style or appearance of the context visual according to the reference image. Following from Eq. 2, the guidance from an energy function can be expressed as follows:

$$\hat{\epsilon}_t = (1 + \omega)\epsilon_\theta(x_t; t, y) - \omega\epsilon_\theta(x_t; t, \emptyset) + \gamma\nabla_{x_t}\mathcal{E}(x_t; t, y), \tag{3}$$

where $\omega$ is the classifier-free guidance strength, and $\gamma$ is the newly introduced guidance weight for the energy function $\mathcal{E}(x_t; t, y)$. In this section, $x_t, \phi, \theta, \emptyset$ represent diffused signal, classifier parameters, noise estimator parameters, null token respectively.

## 4 METHODS

To address the task of *Semantic Style Transfer*, it is essential to establish precise semantic mappings between the content to be transferred and the target. And the style features and visual appearance are required to be transferred with the guidance from semantic correspondence, while preserving the main structure of the original content.

To achieve this, we propose *Semantix*, a novel energy-guided sampler built upon off-the-shelf diffusion models, as illustrated in Fig. 2. Given a context image $I^c$ or video $V^c$ and the reference image $I^{ref}$, we begin with inverting these images or videos to latents $x_T$ through the edit-friendly DDPM inversion (Fig. 2 (left)). In the denoising process, we then modify the target $x_t^{out}$, which is initialized using the inverse noise of the context visual $x_T^c$ at the final step $T$, through the designed energy gradient in every sampling step(Fig. 2 (right)). During the sampling process, we integrate the guidance from our sampler with classifier-free guidance to generate high-quality samples (Liu et al., 2022; Epstein et al., 2023). Notably, our approach only provides additional guidance during sampling, without altering the generative capabilities of the visual model. As a result, the pre-trained video model inherently maintains motion consistency without additional modification. Besides, an AdaIN (Huang and Belongie, 2017) is employed to harmonize color disparities among $I^{out}$ and $I^{ref}$, which is also widely used in recent works (Alaluf et al., 2023; Chung et al., 2023). The algorithm details can be found in Alg. 1.

### 4.1 DDPM INVERSION

To enhance image or video style editing capabilities, we first employ DDPM inversion (Huberman-Spiegelglas et al., 2023) to invert the input to the noise space as outlined in Appendix A. This method significantly reduces the reconstruction errors associated with CFG. As shown in Fig. 2 (left), given a reference image $I^{ref}$ and a context image $I^c$ or video $V^c$, we first derive the respective independent inversion noise sequences $\{\boldsymbol{x}_T^{ref}, \boldsymbol{z}_T^{ref}, \boldsymbol{z}_{T-1}^{ref}, \ldots, \boldsymbol{z}_1^{ref}\}$ for $I^{ref}$, and $\{\boldsymbol{x}_T^c, \boldsymbol{z}_T^c, \boldsymbol{z}_{T-1}^c, \ldots, \boldsymbol{z}_1^c\}$ for $I^c$ or $V^c$ in the forward process. As shown in Fig. 3, at $t = 601$, the features we extracted from the diffusion model contain sufficient contextual information and exhibit precise semantic correspondence between context and reference images. Thus, we revert the images to the $T = 601$ timestep. Once obtained noise sequences, these noise maps are fixed for using in the guided sampling process later. We then initialize $\boldsymbol{x}_T^{out}$, $\boldsymbol{z}_t^{out}$ as $\boldsymbol{x}_T^c$, $\boldsymbol{z}_t^c$ and only manipulate the predictions from this sequence.

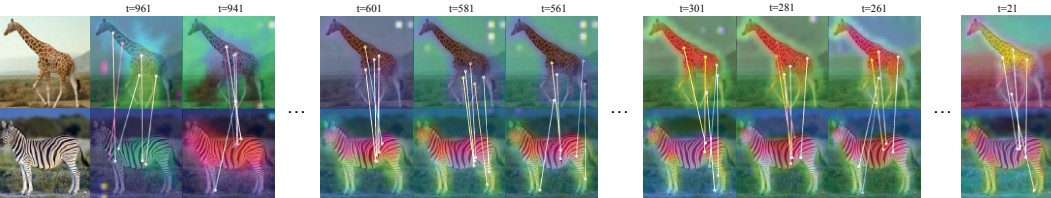

**Figure 3: Visualizing feature maps.** We extracted features from the second block of the diffusion model decoder and visualized the top three PCA components and feature mapping at each timestep.

### 4.2 DESIGN OF ENERGY FUNCTION

Inspired by guidance-based image generation and editing methods (Epstein et al., 2023; Mou et al., 2023), we regulate the sampling process through the design of an energy function to achieve semantic style transfer. By leveraging the gradient of the energy function, we can guide the context visual toward the desired style while preserving its structural integrity. Eq. 3 can be re-expressed as follows:

$$\hat{\epsilon}_t = (1 + \omega)\epsilon_\theta(\boldsymbol{x}_t; t, \mathcal{C}) - \omega\epsilon_\theta(\boldsymbol{x}_t; t, \phi) + \nabla_{\boldsymbol{x}_t}\mathcal{F}(\boldsymbol{x}_t; t, \mathcal{C}), \tag{4}$$

where $\omega$ is the classifier-free guidance strength and $\mathcal{F}$ is the designed energy function which includes three parts: $\mathcal{F}_{ref}$ for Style Feature Guidance, $\mathcal{F}_c$ for Spatial Feature Guidance, and $\mathcal{F}_{reg}$ for regularisation:

$$\mathcal{F}(\boldsymbol{x}_t; t, \mathcal{C}) = \gamma_{ref}\mathcal{F}_{ref} + \gamma_c\mathcal{F}_c + \gamma_{reg}\mathcal{F}_{reg}, \tag{5}$$

where $\gamma$ is the weight of corresponding components. We detail the three components below.

**Style Feature Guidance**  To achieve semantic style transfer, we propose style feature guidance, which accurately captures the semantic correspondence between reference and context features to guide style feature injection. In DIFT (Tang et al., 2023), the authors observed that the internal features of the pre-trained Stable Diffusion model can be used to establish accurate semantic correspondence. Inspired by it, we initially feed the inversion latents $\boldsymbol{x}_t^c, \boldsymbol{x}_t^{ref}$ and the infusion latent $\boldsymbol{x}_t^{out}$ to the diffusion model and acquire the corresponding feature maps $F_t^c, F_t^{ref}$ & $F_t^{out} \in \mathbb{R}^{\{B, c', h', w'\}}$ from the intermediate layers of the network. For videos, feature maps $F_t^c$ & $F_t^{out}$ are extracted from each individual frame $I^c \in V^c, I^{ref} \in V^{ref}$. Next, we pre-align the features via DIFT. Specifically, given the features $F_t^c, F_t^{ref}$ and pixels $p_i, p_j$ in $I^c, I^{ref}$ respectively, we calculate the $\ell_2$ distance between the pairwise vectors $\mathrm{v}_{p_i}^c$ from $F_t^c$ and $\mathrm{v}_{p_j}^{ref}$ from $F_t^{ref}$, and then define the pixels with the smallest distance as the corresponding pixels:

$$D_{ij} = \|\mathrm{v}_{p_i}^c - \mathrm{v}_{p_j}^{ref}\|_2^2, \quad \forall \mathrm{v}_{p_i}^c \in F_t^c, \quad \forall \mathrm{v}_{p_j}^{ref} \in F_t^{ref}, \tag{6}$$

$$p_j^* = \arg\min_{p_j} D_{ij}. \tag{7}$$

Therefore, for any context feature vector of the context image $\mathrm{v}_{p_i}^c \in F_t^c$, the corresponding feature pair can be defined as $\left(\mathrm{v}_{p_i}^c, \mathrm{v}_{p_j}^{ref}\right)$. For all $\mathrm{v}^c$ in $F_t^c$, we can identify corresponding feature pairs. Consequently, we obtain a new set $F_t^{ref*}$ according to corresponding feature pairs.

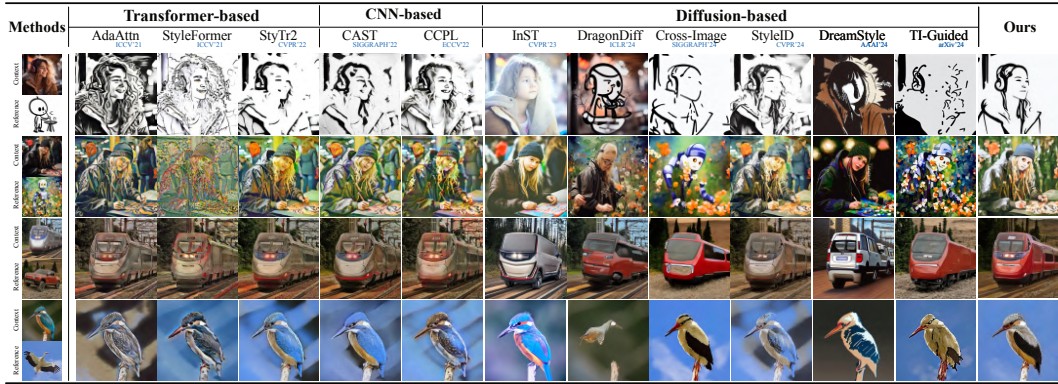

Figure 4: **Qualitative comparison with style transfer and appearance transfer methods.** The top two rows are comparisons of style transfer, the bottom two of appearance transfer.

MCCNet     UNIST     Cross-Image     CCPL     **Ours**     MCCNet     UNIST     Cross-Image     CCPL     **Ours**

Figure 5: **Qualitative comparison of video style transfer.** Click the images to play the animation clips. (Recommended to use Adobe Reader to ensure the GIFs play properly.)

However, such an affinity-based approach overlooks the spatial location of the vectors within the context features and fails to account for their relationships with adjacent vectors. Inspired by Position Encoding (PE) (Vaswani et al., 2017; Dosovitskiy et al., 2020), we integrate an additional optimizing-free position encoding term to maintain relative position. Thus the feature maps can be regarded as:

$$\bar{F}^c_{t_{\{i\}}} \leftarrow F^c_t + \lambda_{pe} \cdot \boldsymbol{pe}_{\{i\}}, \tag{8}$$

$$\bar{F}^{ref}_{t_{\{i\}}} \leftarrow F^{ref}_t + \lambda_{pe} \cdot \boldsymbol{pe}_{\{i\}}. \tag{9}$$

Subsequently, we locate the corresponding pairs of features $\bar{F}^{ref*}_t$ using the $\ell_2$ distance, and utilize the obtained new pairs of feature vectors $\left(\mathrm{v}^c_{p_i}, \mathrm{v}^{ref}_{\bar{p}^*_j}\right)$ to form the new rearranged $F^{ref*}_t$. Therefore, the optimization objective for Style Feature Guidance is to minimize:

$$\mathcal{F}_{ref} \propto \boldsymbol{d}\left(F^{out}_t, F^{ref*}_t\right), \tag{10}$$

where $\boldsymbol{d}$ is a type of distance metric. We use the cosine similarity as implemented in (Mou et al., 2023). Besides, we use self-attention (Tumanyan et al., 2023) to define region masks $m_c$ and $m_{ref}$, which are generated via $k$-means clustering and applied to the features, limiting calculations and guidance to the masked regions.

**Spatial Feature Guidance**    To preserve the spatial structure of the generated content during style guidance, we introduce spatial feature guidance. It minimizes the distance between context and output features, ensuring spatial integrity. Unlike previous methods that replace features, we calculate feature distances at corresponding positions during sampling and design an energy function to align and maintain spatial structure. Specifically, based on the features $F^c_t$ extracted from the context

image or video, we perform point-to-point feature mapping between $F_t^c$ and $F_t^{out}$ to obtain feature vector pairs $(\mathrm{v}_{p_i}^c, \mathrm{v}_{p_i}^{out})$ where $p_i \in F_t^c$. Therefore, we can calculate the similarity between the output features $F_t^{out}$ and the context features $F_t^c$ as Spatial Feature Guidance:

$$\mathcal{F}_c \propto \boldsymbol{d}\left(F_t^{out}, F_t^c\right). \tag{11}$$

**Semantic Distance**    To avoid overfitting to style or context and achieve a balance between style and structure, we incorporate a commonly used regularisation term in training-based methods. Previous methods have demonstrated that self-attention and cross-attention mechanisms encode context and structural information of images (Hertz et al., 2022; Epstein et al., 2023; Tumanyan et al., 2023). Additionally, recent methods have achieved style transfer by swapping the keys and values in self-attention (Alaluf et al., 2023; Chung et al., 2023; Deng et al., 2023; Hertz et al., 2023; Wang et al., 2024b; Jeong et al., 2024). Inspired by these works, we design a regularisation term to balance style injection and context preservation. Specifically, the parameters of style and spatial guidance are their respective feature vectors, while the swapped attention features combine the outputs of both. By constraining and penalizing the parameters from the swapped attention features in the regularization terms, it ensures that the style and spatial feature vectors remain within a more compact space, resulting in a smoother solution, reduced overfitting, and enhanced stability during sampling. During the sampling process, we feed $x_t^{out}$ into U-net again and replace the original $K_l^{out}, V_l^{out}$ in self-attention with $K_l^{ref}, V_l^{ref}$, which come from the reference image $I^{ref}$. Expressed as follows:

$$\text{Self-Attn}(Q_l^{out}, K_l^{ref}, V_l^{ref}) = \text{Softmax}\left(\frac{Q_l^{out}K_l^{ref^T}}{\sqrt{d}}\right)V_l^{ref}. \tag{12}$$

where $l$ is the index of the transformer blocks that need to replace $K$ and $V$, and $d$ is the dimension. Consequently, we calculate the L2 distance between the cross-attention map $\text{Cross-Attn}_{swap}^{out}$ after replacing $K$ and $V$ and the cross-attention map $\text{Cross-Attn}^c$ of the context image:

$$\mathcal{F}_{reg} = \|\text{Cross-Attn}_{swap}^{out} - \text{sg}(\text{Cross-Attn}^c)\|_2^2, \tag{13}$$

where $\text{sg}(\cdot)$ represents the operation of stop gradient.

## 5    EXPERIMENTS

In this section, we conduct an exhaustive experimental analysis to substantiate the efficacy and superiority of our proposed method through qualitative comparison (Sec. 5.1), quantitative comparison (Sec. 5.2) and ablation study (Sec. 5.3). For more experimental details, please refer to the Appendix B.

### 5.1    QUALITATIVE COMPARISON

We compare our proposed method with previous state-of-the-art methods in style and appearance transfer, including Transformer-based methods AdaAttn (Liu et al., 2021), StyleFormer (Wu et al., 2021), StyTr2 (Deng et al., 2022), CNN-based method CAST (Zhang et al., 2022), CCPL (Wu et al., 2022) and Diffusion-based methods InST (Zhang et al., 2023b), Dragon Diffusion (Mou et al., 2023), Cross-Image (Alaluf et al., 2023), StyleID (Chung et al., 2023), DreamStyler (Ahn et al., 2024), TI-Guided-Edit (Wang et al., 2024b). As illustrated in Fig. 4, our approach excels at generating visually appealing images in both style and appearance transfer tasks. Specifically, our method not only preserves the structural integrity of the context image, but also integrates style features more effectively based on semantic correspondence. AdaAttn, Styleformer, StyTr2, CAST and CCPL mainly change the color without injecting

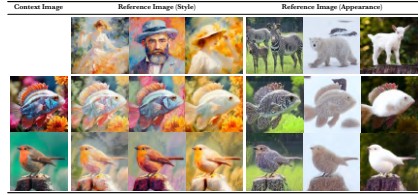

(a) Image Examples w/ Semantix

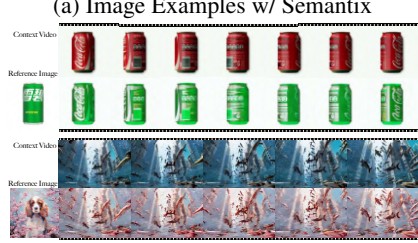

(b) Video Examples w/ Semantix

Figure 6: Image and Video Examples for Semantic Style Transfer

Table 1: Quantitative comparison with style transfer and appearance transfer methods.

| Metrics | AdaAttn | StyleFormer | StyTR2 | CAST | CCPL | InST | Cross-Image | StyleID | DreamStyler | TI-Guided | Ours |
|---|---|---|---|---|---|---|---|---|---|---|---|
| LPIPS ↓ | 0.581 | 0.560 | 0.476 | 0.465 | 0.523 | 0.548 | 0.703 | 0.514 | 0.580 | 0.649 | **0.461** |
| CFSD ↓ | 0.189 | 0.156 | 0.155 | 0.133 | 0.133 | 0.408 | 0.232 | 0.160 | 0.789 | 0.183 | **0.117** |
| SSIM ↑ | 0.403 | 0.331 | 0.561 | 0.514 | 0.536 | 0.383 | 0.454 | 0.527 | 0.334 | 0.453 | **0.589** |
| Gram Metrics$_{\times 10^2}$ ↓ | 7.929 | 2.822 | 5.403 | 6.594 | 4.861 | 4.917 | 5.850 | 2.878 | 6.990 | 4.811 | **2.524** |
| PickScore ↑ | 16.87 | 18.85 | 16.76 | 16.72 | 16.75 | 16.80 | 17.45 | 19.68 | 16.80 | 18.39 | **19.95** |
| HPS ↑ | 16.81 | 18.20 | 16.81 | 16.77 | 16.79 | 16.87 | 16.59 | 18.70 | 16.87 | 17.56 | **18.78** |

The **best** results are highlighted in **bold font**, and the second-best are underlined.
We compare our method with recent state-of-the-art methods in terms of structure preservation, style similarity and image aesthetics.
* To measure structure preservation capability, we calculate the LPIPS, CFSD and SSIM.
* For style similarity, we compute Gram Metrics as style loss.
* We utilize PickScore and HPS as aesthetic evaluation metrics.

the whole style information. InST and DreamStyler fail to effectively learn style and disrupt the context image. Dragon Diffusion employs global feature guidance for appearance transfer, yet fails to maintain structural integrity, causing severe distortion of image context. Meanwhile, Cross-Image leads to significant structural degradation and results in a blending of foreground and background. StyleID is unable to infuse enough style information, potentially leading to color deviation. Furthermore, TI-Guided Edit struggles with semantic-driven style transfer, causing blurry images and structural damage. For video style transfer, we compare our method with previous video style transfer methods such as MCCNet (Deng et al., 2021), UNIST (Gu et al., 2023) and CCPL (Wu et al., 2022). We also adapt Cross-Image (Alaluf et al., 2023) to a video version for comparison. As shown in Fig. 5, our method outperforms others in terms of visual quality, consistency, and continuity. Additional qualitative comparisons of images are provided in the Appendix G. Besides, we provide image and video semantic style transfer results as shown in the Fig. 6, as well as additional visual results in the Appendix I.

## 5.2 QUANTITATIVE COMPARISON

**Metrics.** For quantitative performance of image semantic style transfer, we evaluate stylized images from three aspects: structure preservation ability, style injection capability and image quality. We quantify the structure preservation ability by LPIPS (Zhang et al., 2018), CFSD (Chung et al., 2023) and SSIM (Wang et al., 2004). For style injection capability, we employ Gram matrices, which are widely used in style and appearance transfer (Gatys et al., 2016; Alaluf et al., 2023; Wang et al., 2024b). In addition, we utilize the aesthetic score metrics to measure the quality of the generated images such as PickScore (Kirstain et al., 2024) and Human Preference Score (HPS) (Wu et al., 2023). For video transfer, we evaluate video continuity with Semantic and Object Consistency metrics and a Motion Alignment metric for assessing the difference of motions between source and editing videos as described in (Sun et al., 2024). We further evaluate the visual appeal, motion quality and temporal consistency by Visual Quality, Motion Quality and Temporal Consistency metrics respectively as described in EvalCreafter (Liu et al., 2024).

**Comparison Analysis** We quantitatively evaluate our method on 1000 sampled context-style image pairs and compare it with previous state-of-the-art methods mentioned above. As demonstrated in Tab. 1, our method not only significantly surpasses previous techniques but also excels over recent diffusion-based methods in several context preservation metrics, including LPIPS, CFSD and SSIM. It indicates that our method possesses a significant edge in maintaining structural integrity. For the style similarity

Table 2: Quantitative Comparison of Video Style Transfer.

| Metric | MCCNet | UNIST | Cross-Image | CCPL | Ours |
|---|---|---|---|---|---|
| Semantic Consistency ↑ | 0.714 | 0.861 | 0.936 | 0.942 | **0.944** |
| Object Consistency ↑ | 0.723 | 0.777 | 0.939 | 0.943 | **0.955** |
| Motion Alignment ↑ | -5.251 | -4.178 | -3.878 | **-1.792** | -1.894 |
| Visual Quality ↑ | 52.11 | 43.97 | 47.33 | 48.92 | **55.86** |
| Motion Quality ↑ | 53.35 | **55.07** | 53.14 | 53.25 | 53.99 |
| Temporal Consistency ↑ | 59.14 | 45.43 | 55.85 | 59.64 | **60.05** |

The **best** results are highlighted in **bold font**, and the second-best results are underlined.

metric, we measure the L2 loss between Gram Matrices (Gatys et al., 2015), which represents the style similarity between the generated image and the reference style image. Our Semantix has the lowest Gram Matrices style loss among all methods. This demonstrates that the images generated by our method most closely resemble the reference style images. Besides, Semantix achieves the highest PickScore and HPS, which indicates that the images generated by Semantix are more visually attractive. Therefore, experiments illustrate that Semantix not only solves the problem of context disruption, but also effectively injects style into context images in a harmonious way.

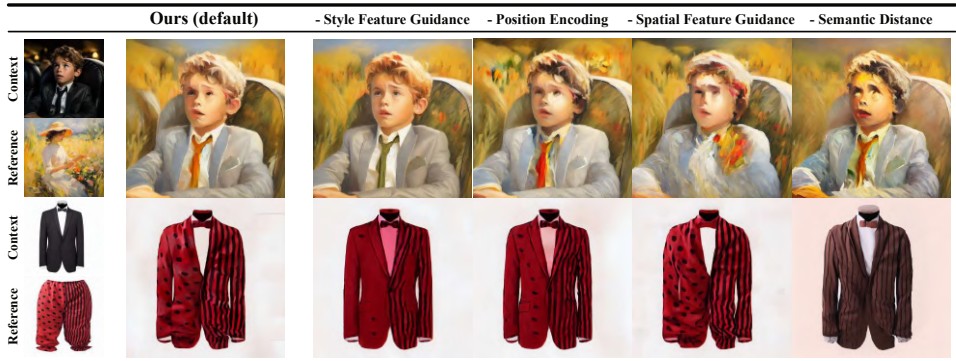

Figure 7: Ablation study on our proposed components. From the 3rd to the 6th columns, each column has a separate component removed compared to our default.

In terms of video semantic style transfer, we conducted evaluations across 100 stylized videos in comparison with MCCNet (Deng et al., 2021), UNIST (Gu et al., 2023), Cross-Image (Alaluf et al., 2023) and CCPL (Wu et al., 2022). In contrast to Cross-Image, our approach did not modify the model's structure, which allowed it to achieve commendable outcomes in generation quality, consistency and continuity aspects. As shown in Tab. 2, our approach exhibited superior performance across all metrics among all methods.

Table 3: User Study.

| | StyleFormer | Cross-Image | StyleID | TI-Guided | Ours |
|---|---|---|---|---|---|
| **Context Preservation** | 8.2% | 3.2% | 33.6% | 3.2% | **51.8%** |
| **Style Similarity** | 3.9% | 21.8% | 15.0% | 28.2% | **31.1%** |
| **Visual Appeal** | 7.8 % | 3.6% | 32.1% | 5.4% | **51.1%** |

* We ask all volunteers to evaluate the stylized images based on three aspects: context preservation, style similarity, and visual appeal.
* The results are averaged across all volunteers

Table 4: Ablation study on our proposed components.

| Metric | Ours(default) | - Style Guidance | - Spatial Guidance | - Position Encoding | - Semantic Distance |
|---|---|---|---|---|---|
| **LPIPS** ↓ | 0.461 | 0.406 | 0.512 | 0.467 | 0.451 |
| **CFSD** ↓ | 0.117 | 0.116 | 0.154 | 0.136 | 0.121 |
| **SSIM** ↑ | 0.589 | 0.610 | 0.547 | 0.574 | 0.626 |
| **Gram Metrics**$_{\times 10^2}$ ↓ | 2.5242 | 4.2289 | 2.6315 | 2.8982 | 2.8610 |
| **PickScore** ↑ | 19.945 | 16.756 | 16.756 | 16.758 | 16.758 |
| **HPS** ↑ | 18.7801 | 16.769 | 16.768 | 16.769 | 16.766 |

From the 3rd to the 6th columns, each column has a separate component removed compared to the second column (ours default).

**User Study** In order to obtain a more convincing comparison, we conducted a user study to compare our method with StyleFormer (Wu et al., 2021), Cross-Image (Alaluf et al., 2023), StyleID (Chung et al., 2023) and TI-Guided (Wang et al., 2024b). We enrolled 30 volunteers, and for each volunteer we randomly selected 40 generated images for user evaluation in three aspects: structural preservation, style similarity and visual appeal. The results, as detailed in the Tab. 3, demonstrate that our method excels at maintaining image structure, enhancing style injection, and achieves superior image quality.

### 5.3 ABLATION STUDY

To validate the effects of each component, we conducted a series of ablation studies. In particular, we experiment with style feature guidance, spatial feature guidance, position encoding and semantic distance in both qualitative and quantitative aspects. As illustrated in the Tab. 4 and Fig. 7, style feature guidance injects style features into context visual based on semantic correspondence. Spatial feature guidance effectively maintains the structure, thereby ensuring the coherence of image context. Position Encoding enhances the harmony of stylized images. As a regularisation term, semantic distance mitigates overfitting to structure or style and contributes to balancing style injection and structure maintenance. Further analysis is provided in Appendix E. Parameter sensitivity analysis and ablation on semantic distance can be found in Fig. 13 and Fig. 15 respectively.

## 6 CONCLUSION

We propose *Semantix*, a carefully crafted energy-guided sampler designed to facilitate semantic alignment feature transfer in both image and video diffusion models. By utilizing three proposed components, Semantix effectively transferred both the style and appearance features with semantic guidance. Experimental results demonstrate that our approach not only produces high-quality stylized images and videos but also efficiently prevents contextual interference and effectively incorporates style features. Moreover, video experiments confirm that our method excels in maintaining consistency and continuity across frames.

## LIMITATIONS

While our method achieves style and appearance transfer across various visual, there are still certain limitations. Specifically, it tends to be less effective when the context visual has strong inherent style features. In such scenarios, our method may lead to failures in semantic style transfer. Moreover, Semantix relies on the semantic information from pre-trained diffusion models, with the UNet architecture (Rombach et al., 2022). Future work will explore semantic alignment in advanced models like UViT (Bao et al., 2023) and DiT (Peebles and Xie, 2023).

## BROADER IMPACTS

Powerful feature transfer capabilities can facilitate creative content generation. However, there is also a risk that these technologies could be used to generate harmful content with negative societal impacts. A particularly concerning misuse could involve placing an actual people in a compromising scene involving illicit activities.

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

# Appendix for Semantix: An Energy Guided Sampler for Semantic Style Transfer

## A    EDIT FRIENDLY INVERSION

This section reproduces, for the reader's convenience, much of the derivation for edit friendly inversion presented in Huberman-Spiegelglas et al. (2023). Diffusion models (Sohl-Dickstein et al., 2015) consist of a forward process and a denoise process. In the forward process, an image $x_0$ is transformed into a Gaussian distribution $x_T$ by gradually adding Gaussian noise. This process can be described by a stochastic differential equation (SDE) (Song et al., 2020b):

$$\mathrm{d}x = f(\mathbf{x}, t)\,\mathrm{d}t + g(t)\,\mathrm{d}w, \tag{14}$$

here, $w$ represents the standard Wiener process, $f(\cdot, t) : \mathbb{R}^d \to \mathbb{R}^d$ is a vector-valued function known as the drift coefficient of $x(t)$, and $g(\cdot) : \mathbb{R} \to \mathbb{R}$ is a scalar function referred to as the diffusion coefficient of $x(t)$. In DDPM (Ho et al., 2020), the SDE can be discretized into the following form:

$$x_t = \sqrt{1 - \beta_t}x_{t-1} + \sqrt{\beta_t}\epsilon_t, \quad t = 1, \ldots, T \tag{15}$$

where $\epsilon_t \sim \mathcal{N}(0, I)$ represents standard Gaussian noise, and $\beta_t$ is variance schedule. Formally, the diffusion process can be equivalently expressed as:

$$x_t = \sqrt{\bar{\alpha}_t}x_0 + \sqrt{1 - \bar{\alpha}_t}\epsilon_t \tag{16}$$

where $\alpha_t = 1 - \beta_t$, $\bar{\alpha}_t = \prod_{s=1}^{t} \alpha_s$, and $\epsilon_t \sim \mathcal{N}(0, I)$. In the denoising process, a Gaussian noise $x_T \sim \mathcal{N}(0, I)$ is progressively denoised to sample an image $\hat{x}_0$. A non-Markovian sampling method was proposed in DDIM (Song et al., 2020a), which can be formulated as follows:

$$x_{t-1} = \hat{\mu}_t(x_t) + \sigma_t z_t, \quad t = T, \ldots, 1 \tag{17}$$

where $z_t \sim \mathcal{N}(0, I)$ represents standard Gaussian noise and is independent of $x_t$. The term $\hat{\mu}_t(x_t)$ can be expressed by the following formula:

$$\hat{\mu}_t(x_t) = \sqrt{\bar{\alpha}_{t-1}}\frac{x_t - \sqrt{1 - \bar{\alpha}_t}\epsilon_\theta(x_t)}{\sqrt{\bar{\alpha}_t}} + \sqrt{1 - \bar{\alpha}_{t-1} - \sigma_t^2}\epsilon_\theta(x_t) \tag{18}$$

the fixed schedulers $\sigma_t$ are represented in the general form as: $\sigma_t = \frac{\eta\beta_t(1 - \bar{\alpha}_{t-1})}{1 - \bar{\alpha}_t}$, where $\eta$ belongs to [0,1]. Within this framework, $\eta = 0$ corresponds to the deterministic DDIM scheme, and $\eta = 1$ corresponds to the original DDPM scheme. Owing to the deterministic nature of the Ordinary Differential Equations (ODEs), the DDIM does not introduce randomness during the diffusion process. However, classifier-free guidance amplifies the accumulation of errors in text-guided sampling process (Mokady et al., 2023), hence the deterministic DDPMs (Huberman-Spiegelglas et al., 2023; Wu and De la Torre, 2023) are proposed. In DDPM inversion, a sequence $x_1, \ldots, x_T$ is constructed directly from $x_0$ according to the following formula:

$$x_t = \sqrt{\bar{\alpha}_t}x_0 + \sqrt{1 - \bar{\alpha}_t}\tilde{\epsilon}_t, \quad t = 1, \ldots, T, \tag{19}$$

where $\tilde{\epsilon}_t \sim \mathcal{N}(0, I)$ are statistically independent, which are different from Eq. 15. Therefore, we can compute $\hat{\mu}_t(x_t)$ according to the Eq. 18, which allows us to isolate $z_t$ according to Eq. 17 as shown in the following equation:

$$z_t = \frac{x_{t-1} - \hat{\mu}_t(x_t)}{\sigma_t}, \quad t = T, \ldots, 1. \tag{20}$$

Therefore, the diffusion process evolves into a deterministic DDPM. It has been demonstrated that the introduction of randomness in image editing tasks can produce high quality results (Nie et al., 2023).

## B    IMPLEMENTATION DETAIL

We use NVIDIA A100 (80G) GPUs for all experiments. For image semantic style transfer, our method is built upon the pre-trained Stable Diffusion v1.5 model. For video task, AnimateDiff (Guo et al., 2023) serves as our base model and we extend the reference image into video sequence. We invert the input images or videos into noises through DDPM inversion across 60 timesteps. For

classifier-free guidance, we set the scale factor $\omega = 3.5$, aligning it with the sampling procedures. During the sampling process, the features for guidance are extracted from the second and third blocks of the UNet's decoder. In image style transfer tasks, we adjust the weights of style feature guidance, spatial feature guidance and semantic distance regularisation to $\gamma_{ref} = 3.0, \gamma_c = 0.9, \gamma_{reg} = 1.0$, respectively. Additionally, we incorporate a 2D position encoding into the features and assign it a weight of $\lambda_{pe} = 3.0$. For video task, the corresponding hyper-parameters are set to $\gamma_{ref} = 6.0, \gamma_c = 3.0, \gamma_{reg} = 5.0, \lambda_{pe} = 3.0$. We further employ a hard clamp in the range of $[-1, 1]$ for all guidance.

To compute the regularisation term, we extract cross-attention maps of context image or video and output image or video in the U-net to calculate L2 distance. Besides, we replace the keys and values in the self-attention layer of the decoder at the $32 \times 32$ and the $64 \times 64$ resolutions following 10 timesteps, as described in Cross-Image (Alaluf et al., 2023). After 20 denoising timesteps, we apply AdaIN (Huang and Belongie, 2017) for the style latents $\boldsymbol{x}_t^{ref}$ and output latents $\boldsymbol{x}_t^{out}$.

## C  EVALUATION DATASET

We select the *COCO* (Lin et al., 2014) dataset as the source of context images and obtain style images from *WikiArt* (Tan et al., 2018) and appearance images from Cross-Image (Alaluf et al., 2023). We randomly sample 1000 context images in the *COCO* dataset and pair each with a style image from the *WikiArt* dataset to form 1000 context-style image pairs. All images are cropped and resized to $512 \times 512$ resolutions. For context video datas, we obtain high quality videos generated by Sora (OpenAI, 2024). Additionally, we also utilize some of our internal data as style images for qualitative result.

## D  ALGORITHM OF SEMANTIX

In order to make it easier to understand our proposed *Semantix*, we present the detailed algorithm in Algorithm 1.

## E  FURTHER ANALYSIS.

**Sampling speed and memory analysis.**   Since our sampler is guided by energy gradients, there is a slight increase in sampling time and memory usage. Tab. 5 shows a comparison between our method and other diffusion-based approaches, indicating that our approach has a negligible impact on inference time and memory usage.

Table 5: Comparison of computational speed and memory usage.

| Attribute | Time (sec) | Memory (G) |
|---|---|---|
| **DragonDiff** | 15 | 8 |
| **Cross-Image** | 18 | 12 |
| **StyleID** | 6 | 22 |
| **TI-Guided** | 16 | 20 |
| **Ours** | 26 | 22 |

* The evaluations were carried out on a single NVIDIA A100 GPU, with each method executing 50 timesteps of sampling.

**Impact of correspondence accuracy.**   To further validate the versatility of our approach, we randomly disrupted the feature correspondences between context and reference images. As illustrated in the Fig. 9, this random disruption of feature correspondences led to only a slight decline in image quality, without significantly degrading the stylized images. This is attributable to the robustness of our proposed method. Our energy-based approach does not demand high accuracy in semantic correspondences, and the replacement of the K, V in self-attention with semantic correspondences in our semantic distance term also reduces the method's reliance on precise semantic correspondence.

**Choices of Position Encoding.**   In video style transfer, we conducted experiments on 2D and 3D Position Encoding(PE) to explore the impact of different types of PE. The results for 3D PE, as shown in the Fig. 10, illustrate a slight decrease in style similarity compared to 2D PE(Fig. 5). Notably, the role of PE is mainly to establish semantic correspondence. Thus, this decline can be attributed to the increased difficulty in establishing semantic correspondence across all frames, as opposed to within individual frames. Therefore, in video tasks, we employ 2D PE, aligning with image style transfer.

**The selection of timestep.**   In our proposed method, we reverse the input to T=601 timestep by DDPM inversion (Huberman-Spiegelglas et al., 2023). During the experiment, we observed that when T = 601, there was an optimal trade-off between generation quality and sampling speed. Both previous studies (Voynov et al., 2023b; Agarwal et al., 2023a;b) and Fig. 3 indicate that the image

---

**Algorithm 1:** Proposed Semantix

---

**Input:** context image $I^c$ or video $V^c$; reference image $I^{ref}$ or video $V^{ref}$; prompts $P$

**Output:** reconstructed and stylized images/videos $\hat{I}^c(\hat{V}^c)$, $\hat{I}^{ref}(\hat{V}^{ref})$, $\hat{I}^{out}(\hat{V}^{out})$

**Require:** UNet denoiser $\epsilon_{\boldsymbol{\theta}}$; UNet feature extractor $\mathcal{F}_{\boldsymbol{\theta}}$; hyper-parameters $\gamma_{ref}$, $\gamma_c$, $\gamma_{reg}$, $\lambda_{pe}$, $\omega$; timestep $T$

**Initialization:**

(1) Encode image/video:

$\quad \boldsymbol{x}_0^c = \text{Encoder}(I^c/V^c)$;

$\quad \boldsymbol{x}_0^{ref} = \text{Encoder}(I^{ref}/V^{ref})$;

(2) DDPM inversion to obtain latent $\boldsymbol{x}_T$ and noise $\boldsymbol{z}$:

$\quad \boldsymbol{x}_T^c, \boldsymbol{z}_T^c, \boldsymbol{z}_{T-1}^c, \ldots, \boldsymbol{z}_1^c \quad \longleftarrow \quad \text{DDPM inversion}(\boldsymbol{x}_0^c)$;

$\quad \boldsymbol{x}_T^{ref}, \boldsymbol{z}_T^{ref}, \boldsymbol{z}_{T-1}^{ref}, \ldots, \boldsymbol{z}_1^{ref} \quad \longleftarrow \quad \text{DDPM inversion}(\boldsymbol{x}_0^{ref})$;

$\quad \boldsymbol{x}_T^{out}, \boldsymbol{z}_T^{out}, \boldsymbol{z}_{T-1}^{out}, \ldots, \boldsymbol{z}_1^{out} \quad \longleftarrow \quad \boldsymbol{x}_T^c, \boldsymbol{z}_T^c, \boldsymbol{z}_{T-1}^c, \ldots, \boldsymbol{z}_1^c$;

**for** $t = T, \ldots, 1$ **do**

$\quad \boldsymbol{x}_t \longleftarrow \text{Concat}(\boldsymbol{x}_t^c, \boldsymbol{x}_t^{ref}, \boldsymbol{x}_t^{out})$;

$\quad \boldsymbol{\epsilon}_t^c, \boldsymbol{\epsilon}_t^{uc}, \text{CA}^c \longleftarrow \epsilon_{\boldsymbol{\theta}}(\boldsymbol{x}_t; t, P)$;

$\quad \text{CA}_{swap}^{out} \longleftarrow \epsilon_{\boldsymbol{\theta}}(\boldsymbol{x}_t; t, P)$; where $K_l^{out} = K_l^{ref}, V_l^{out} = V_l^{ref}$

$\quad m_c, m_{ref} \longleftarrow \text{Self-Attn}$;

$\quad \hat{\boldsymbol{\epsilon}}_t = \boldsymbol{\epsilon}_t^{uc} + \omega \cdot (\boldsymbol{\epsilon}_t^c - \boldsymbol{\epsilon}_t^{uc})$;

$\quad F_t^c, F_t^{ref}, F_t^{out} = \mathcal{F}_{\boldsymbol{\theta}}(\boldsymbol{x}_t; t, P)$;

$\quad \bar{F}_t^{ref*} = \text{Align}\Big( (F_t^c + \lambda_{pe} \cdot PE)[m_c], (F_t^{ref} + \lambda_{pe} \cdot PE)[m_{ref}] \Big)$;

$\quad \mathcal{F} = \gamma_{ref} \mathcal{F}_{ref}(F_t^{out}, \bar{F}_t^{ref*}) + \gamma_c \mathcal{F}_c(F_t^{out}, F_t^c) + \gamma_{reg} \mathcal{F}_{reg}(\text{CA}_{swap}^{out}, \text{CA}^c)$;

$\quad \hat{\boldsymbol{\epsilon}}_t = \hat{\boldsymbol{\epsilon}}_t + \nabla_{\boldsymbol{x}_t^{out}} \mathcal{F}$;

$\quad \boldsymbol{x}_{t-1} = \sqrt{\bar{\alpha}_{t-1}}(\frac{\boldsymbol{x}_t - \sqrt{1-\bar{\alpha}_t}\hat{\boldsymbol{\epsilon}}_t}{\sqrt{\bar{\alpha}_t}} + \sqrt{1 - \bar{\alpha}_{t-1} - \sigma_t^2}\hat{\boldsymbol{\epsilon}}_t) + \sigma_t \boldsymbol{z}_t$;

$\quad \boldsymbol{x}_{t-1}^{out} = \text{AdaIN}(\boldsymbol{x}_{t-1}^{out}, \boldsymbol{x}_{t-1}^{ref})$;

**end**

Return $\hat{I}^c(\hat{V}^c), \hat{I}^{ref}(\hat{V}^{ref}), \hat{I}^{out}(\hat{V}^{out}) = \text{Decoder}(\boldsymbol{x}_0^c, \boldsymbol{x}_0^s, \boldsymbol{x}_0^{out})$;

---

structure is formed early in the sampling process. Therefore, when T > 601, the spatial structure of the image has not fully developed, and the semantic correspondence is ambiguous. At this stage, feature guidance can disrupt the image structure, thereby reducing the quality of the generated image. When T < 601, the correspondence tends to be more accurate. However, too few sampling steps weaken the guidance of energy function, reducing style transfer effectiveness. Fewer timesteps with more sampling steps may lead to overfitting due to minimal feature variation. Furthermore, we conducted an ablation study on the sampling timesteps, with the results shown in Fig. 8.

## F    QUANTITATIVE COMPARISON IN APPEARANCE TRANSFER.

To quantitatively confirm the effectiveness of our method in the appearance transfer task, we conduct additional evaluations, similar to our quantitative comparisons of style transfer. We compare our method with some other appearance transfer methods (Cross-Image (Alaluf et al., 2023), TI-Guided (Wang et al., 2024b)) on the data sourced from Cross-Image. Our evaluation metrics include structure preservation ability (LPIPS, CFSD, SSIM), style similarity (Gram matrices) and aesthetic scores (PickScore, HPS) on the images of 11 domains provided by Cross-Image. We display the evaluation results of structural preservation capability in Tab. 6,

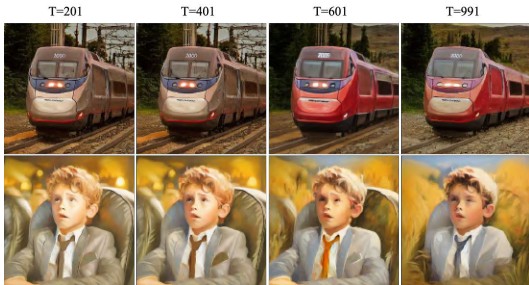

Figure 8: Ablation of sampling timesteps.

and the results of style similarity and aesthetic scores in Tab. 7. It can be seen that our method not only excels in structural preservation but also leads in appearance transfer capability and overall image aesthetics, significantly outperforming competing methods.

Table 6: Structure Preservation Capacity Quantitative Comparisons to Appearance Transfer Methods.

| Structure Preservation | LPIPS ↓ | | | CFSD ↓ | | | SSIM ↑ | | |
|---|---|---|---|---|---|---|---|---|---|
| Domain | Cross-Image | TI-Guided | **Ours** | Cross-Image | TI-Guided | **Ours** | Cross-Image | TI-Guided | **Ours** |
| Animals | 0.6389 | 0.5586 | 0.4340 | 1.0949 | 0.4313 | 0.3659 | 0.3996 | 0.4670 | 0.5347 |
| Birds | 0.5298 | 0.4718 | 0.3920 | 0.4652 | 0.1119 | 0.0753 | 0.6343 | 0.6763 | 0.7283 |
| Buildings | 0.4860 | 0.4001 | 0.3004 | 1.1637 | 0.5102 | 0.4950 | 0.4646 | 0.4644 | 0.5788 |
| Cake | 0.5737 | 0.5274 | 0.3929 | 0.5588 | 0.4225 | 0.2431 | 0.5099 | 0.5235 | 0.6120 |
| Cars | 0.5451 | 0.4794 | 0.3694 | 0.3505 | 0.1673 | 0.1202 | 0.4930 | 0.5012 | 0.6319 |
| Fish | 0.4850 | 0.3995 | 0.3905 | 0.5391 | 0.2234 | 0.2076 | 0.4560 | 0.5131 | 0.5373 |
| Food | 0.4814 | 0.2924 | 0.2924 | 0.4314 | 0.1112 | 0.1112 | 0.5405 | 0.5228 | 0.6905 |
| Fruits | 0.5417 | 0.5062 | 0.3872 | 0.1657 | 0.1391 | 0.0468 | 0.5498 | 0.5322 | 0.6711 |
| House | 0.5268 | 0.4463 | 0.4135 | 0.7968 | 0.4772 | 0.1886 | 0.4023 | 0.3964 | 0.5433 |
| Landscapes | 0.6669 | 0.5893 | 0.4817 | 0.2274 | 0.1578 | 0.0618 | 0.4117 | 0.4186 | 0.5284 |
| Vehicles | 0.5746 | 0.5233 | 0.3990 | 0.7471 | 0.3260 | 0.3474 | 0.4118 | 0.3724 | 0.5564 |
| Average | 0.5500 | 0.4722 | **0.3866** | 0.5946 | 0.2798 | **0.2057** | 0.4794 | 0.4898 | **0.6012** |

Table 7: Appearance Similarity and Aesthetics Scores Quantitative Comparisons to Appearance Transfer Methods.

| Appearance & Aesthetics | Gram Metrices$_{\times 10^2}$ ↓ | | | PickScore ↑ | | | HPS ↑ | | |
|---|---|---|---|---|---|---|---|---|---|
| Domain | Cross-Image | TI-Guided | **Ours** | Cross-Image | TI-Guided | **Ours** | Cross-Image | TI-Guided | **Ours** |
| Animals | 13.4105 | 10.9154 | 7.3016 | 19.13 | 20.29 | 20.56 | 18.99 | 20.18 | 20.27 |
| Birds | 9.1299 | 2.4086 | 1.3395 | 19.27 | 19.56 | 20.20 | 19.53 | 19.81 | 20.31 |
| Buildings | 15.8958 | 14.6079 | 9.9048 | 19.63 | 19.99 | 20.48 | 18.19 | 18.61 | 19.06 |
| Cake | 10.7525 | 11.8198 | 8.0294 | 19.43 | 19.91 | 20.05 | 19.19 | 19.71 | 19.95 |
| Cars | 5.9848 | 3.6776 | 2.6616 | 19.79 | 20.08 | 20.29 | 19.77 | 19.93 | 20.01 |
| Fish | 9.2379 | 11.8551 | 4.6625 | 20.30 | 20.87 | 20.86 | 19.88 | 20.16 | 20.12 |
| Food | 13.2022 | 5.4441 | 2.8371 | 20.71 | 21.67 | 22.26 | 19.62 | 20.27 | 20.68 |
| Fruits | 5.2102 | 3.6503 | 1.2086 | 19.63 | 20.15 | 20.52 | 19.42 | 19.58 | 19.91 |
| House | 16.2923 | 12.8945 | 3.8237 | 20.61 | 20.72 | 20.70 | 19.28 | 19.61 | 19.44 |
| Landscapes | 10.4906 | 7.1602 | 2.1185 | 19.43 | 19.89 | 20.35 | 18.34 | 18.75 | 18.86 |
| Vehicles | 11.1199 | 7.1911 | 6.9143 | 20.04 | 20.59 | 21.16 | 19.25 | 19.65 | 20.16 |
| Average | 10.9751 | 8.3295 | **4.6183** | 19.82 | 20.34 | **20.68** | 19.22 | 19.66 | **19.89** |

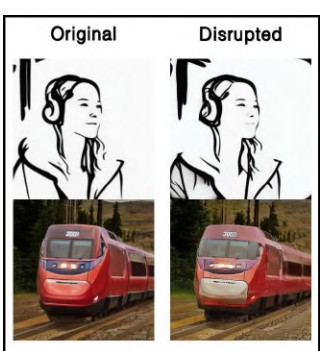

Figure 9: Impact of correspondence accuracy.

Figure 10: Video results with 3D Position Encoding.

## G    ADDITIONAL QUALITY COMPARISON RESULTS.

We provide additional style and appearance transfer qualitative comparison results with diffusion-based methods (Dragondiff (Mou et al., 2023), Cross-Image (Alaluf et al., 2023), StyleID (Chung et al., 2023) and TI-Guided (Wang et al., 2024b)) in Fig. 11 and Fig. 12.

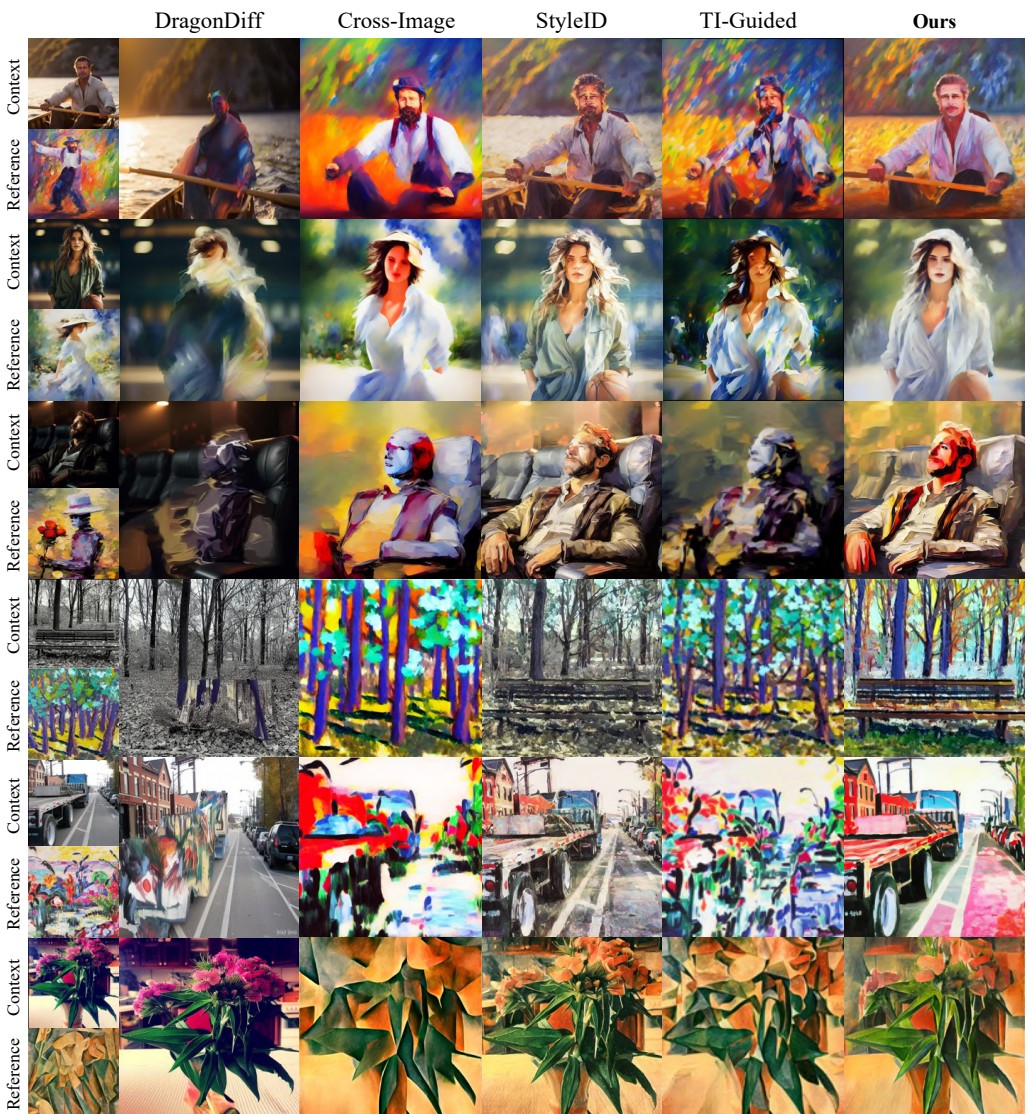

Figure 11: Additional qualitative comparison with baselines (Dragondiff, Cross-Image, StyleID, TI-Guided) on semantic style transfer (style) task.

## H  PARAMETER SENSITIVITY

We further demonstrate the parameter effects on different levels in Fig. 13. We found that excessively high values of $\gamma_{ref}$ can disrupt the structure, while excessively high values of $\gamma_c$ can diminish the impact of style and appearance. The parameter $\gamma_{reg}$ balances $\gamma_{ref}$ and $\gamma_c$, thereby enhancing image quality. Meanwhile, $\lambda_{pe}$ controls the intensity of the affinity between semantic information and the position information. Besides, it also shows that the performance of our proposed sampler is not highly sensitive to these hyper-parameters.

## I  ADDITIONAL VISUAL RESULTS.

We also provide additional image style transfer results in Fig. 14, Fig. 16, and additional image appearance transfer results in Fig. 17 and Fig. 18. Additional video style and appearance transfer results are shown in Fig. 19. For more video results, please refer to the supplementary materials in the submitted file.

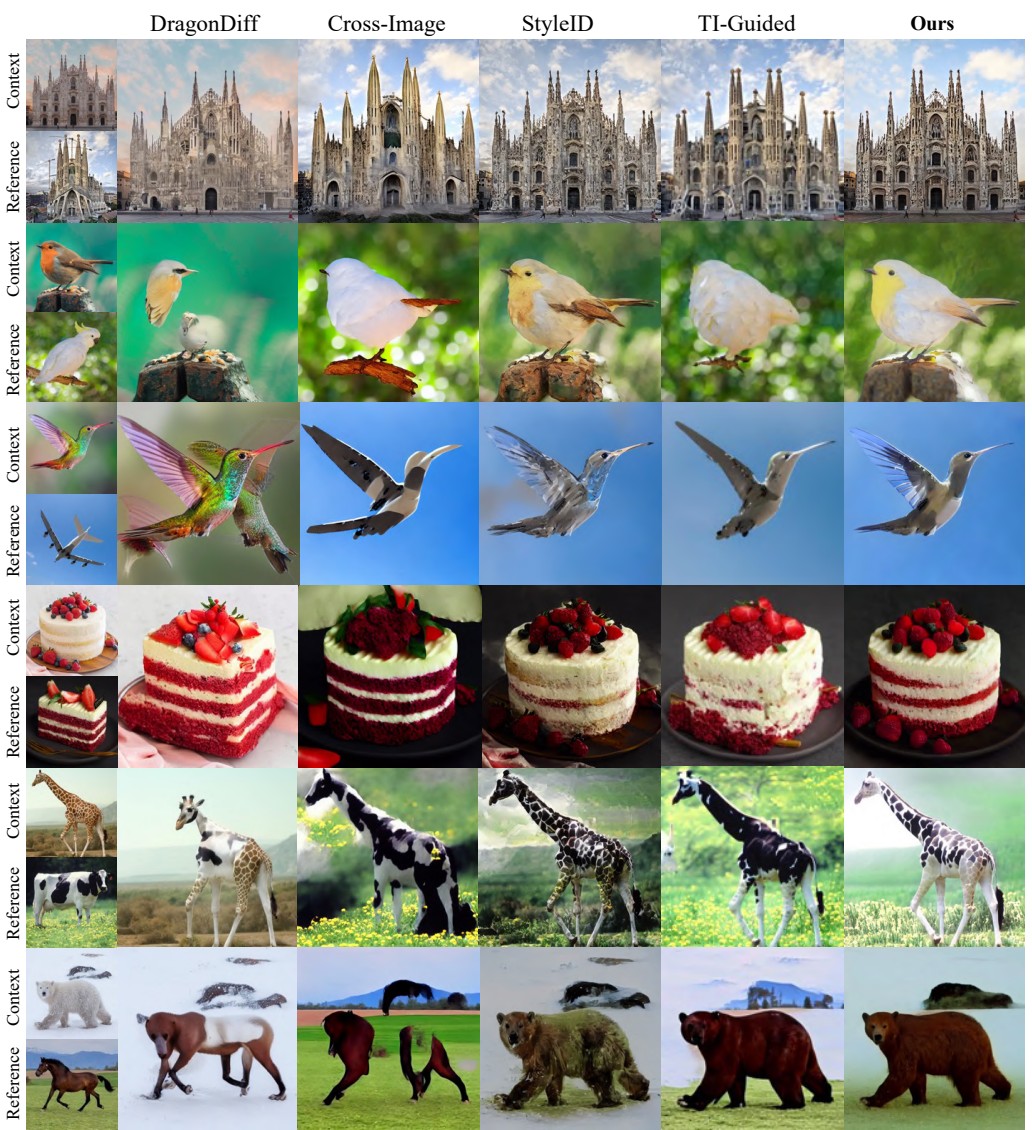

Figure 12: Additional qualitative comparison with baselines (Dragondiff, Cross-Image, StyleID, TI-Guided) on semantic style transfer (appearance) task.

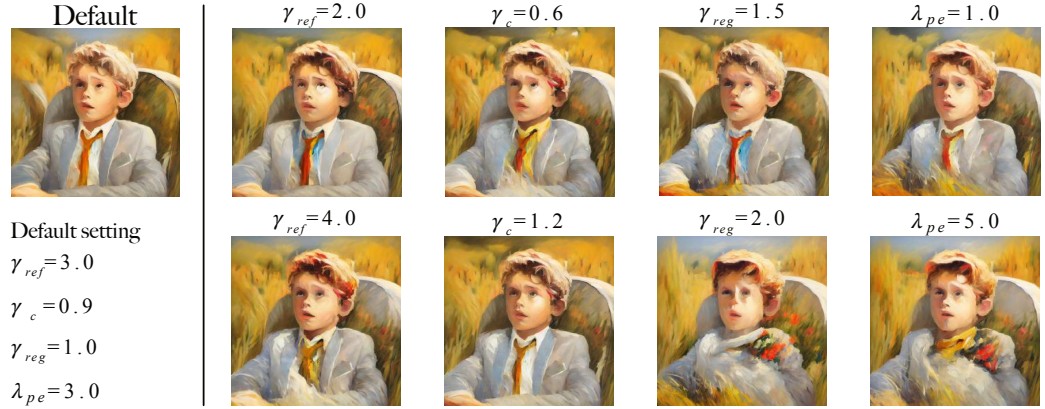

Figure 13: Parameter sensitivity of $\gamma_{ref}$, $\gamma_c$, $\gamma_{reg}$ and $\lambda_{pe}$.

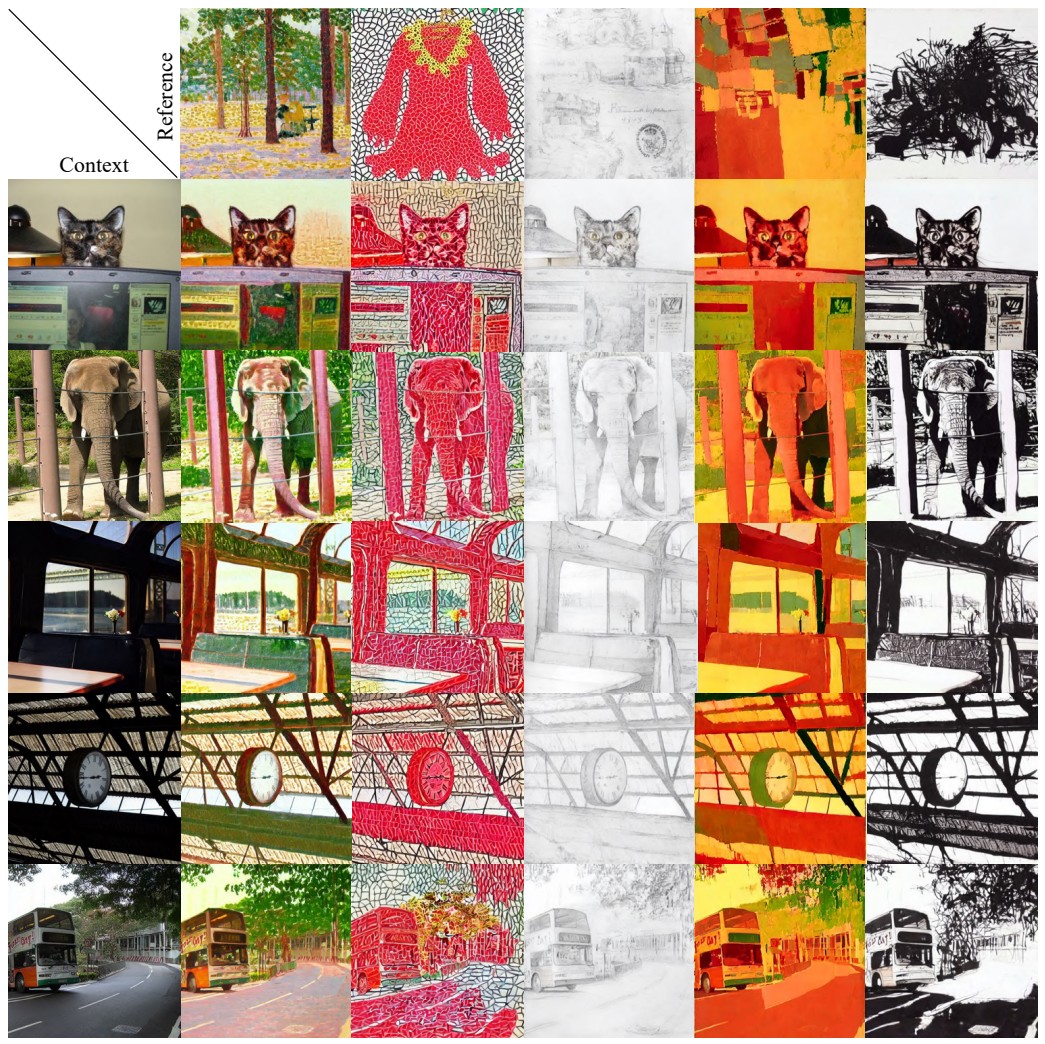

Figure 14: Additional image semantic style transfer (style) results on given context and reference image pairs.

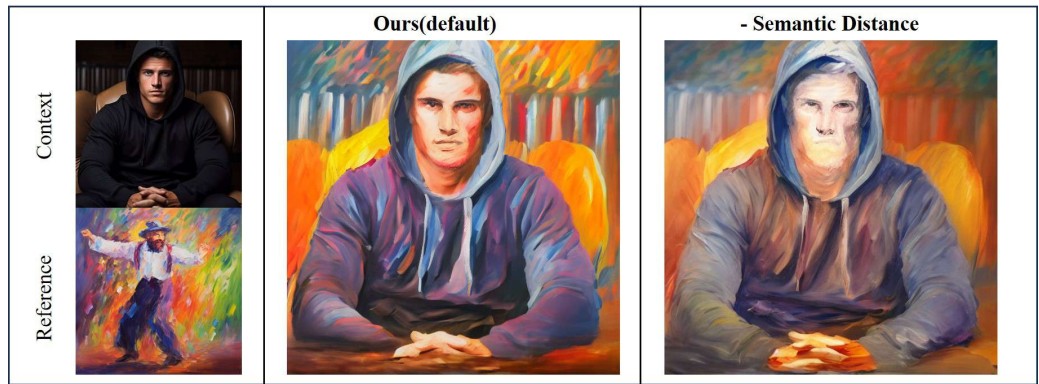

Figure 15: Additional ablation study on Semantic Distance.

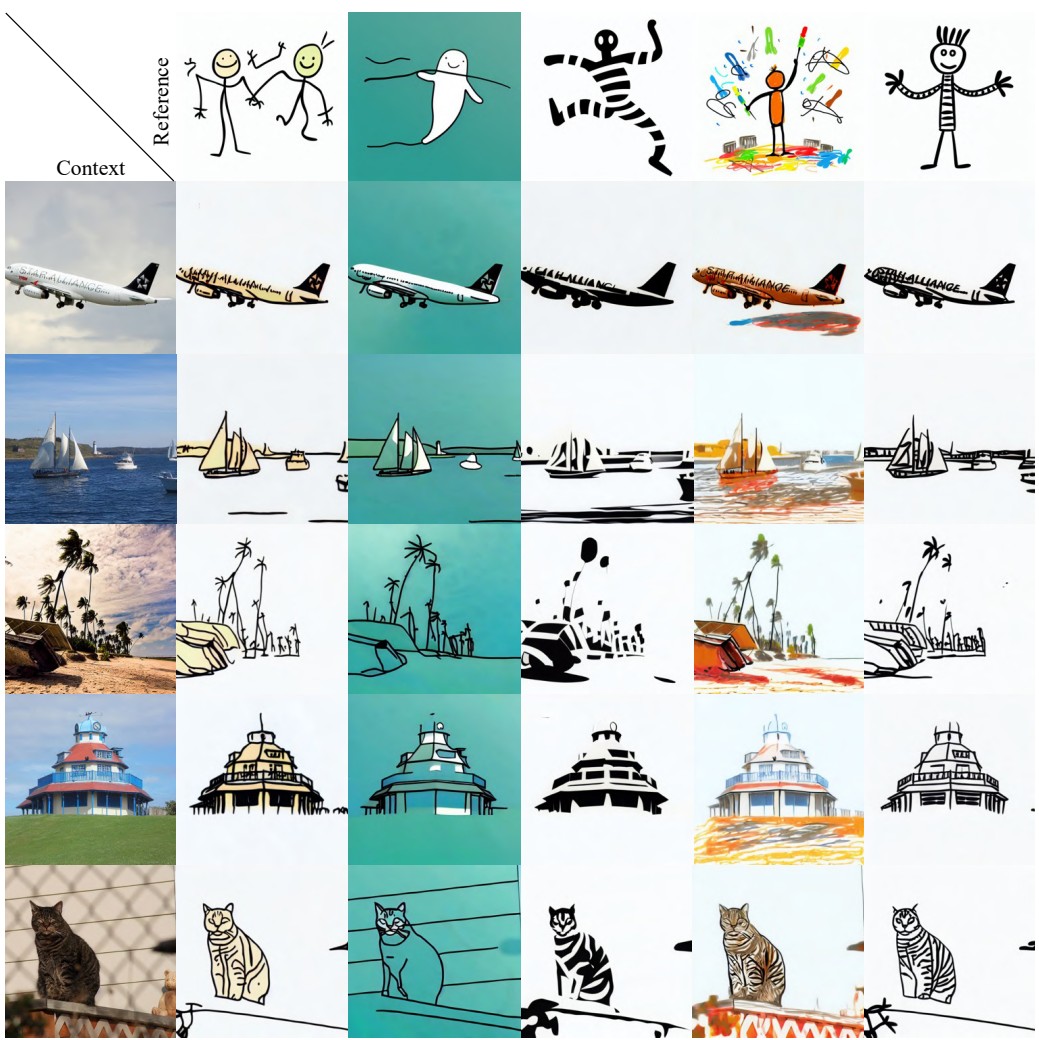

Figure 16: Additional image semantic style transfer (style) results on given context and reference image pairs.

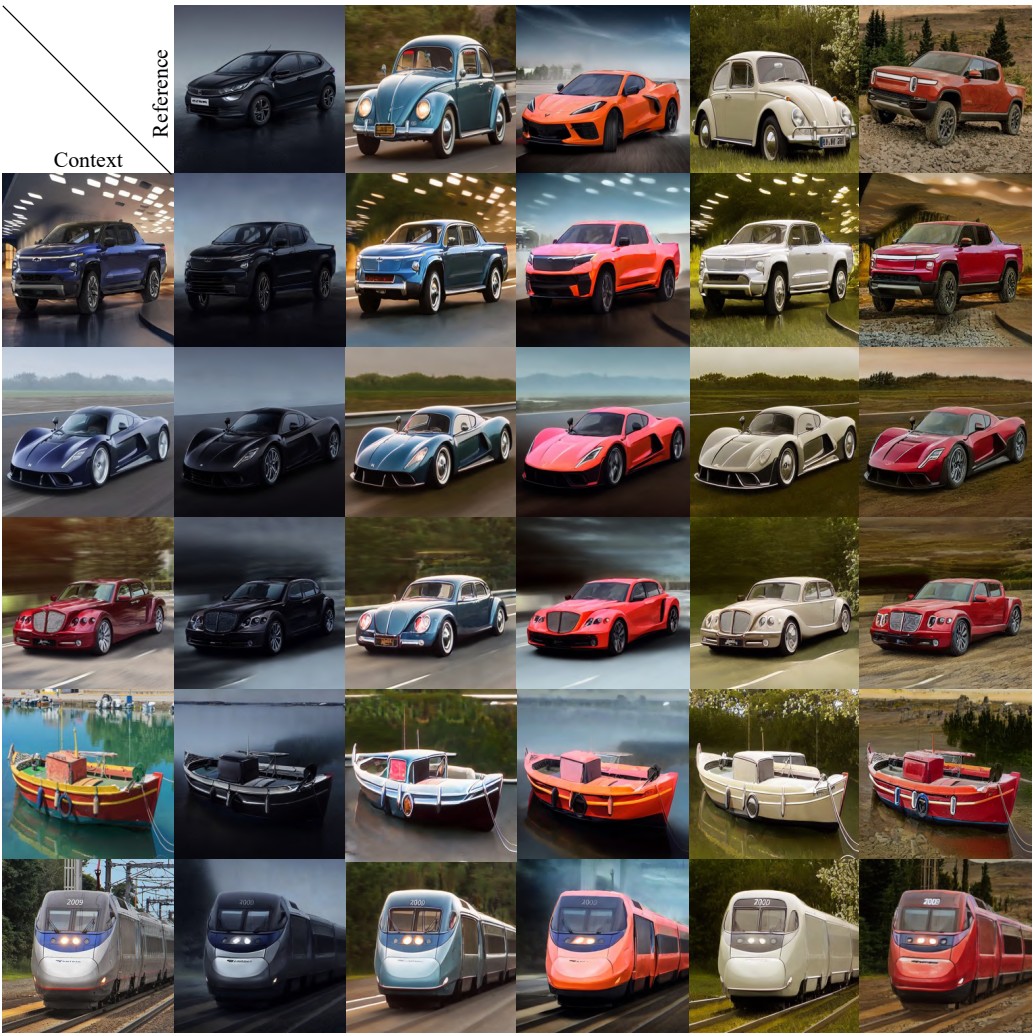

Figure 17: Additional image semantic style transfer (appearance) results on given context and reference image pairs.

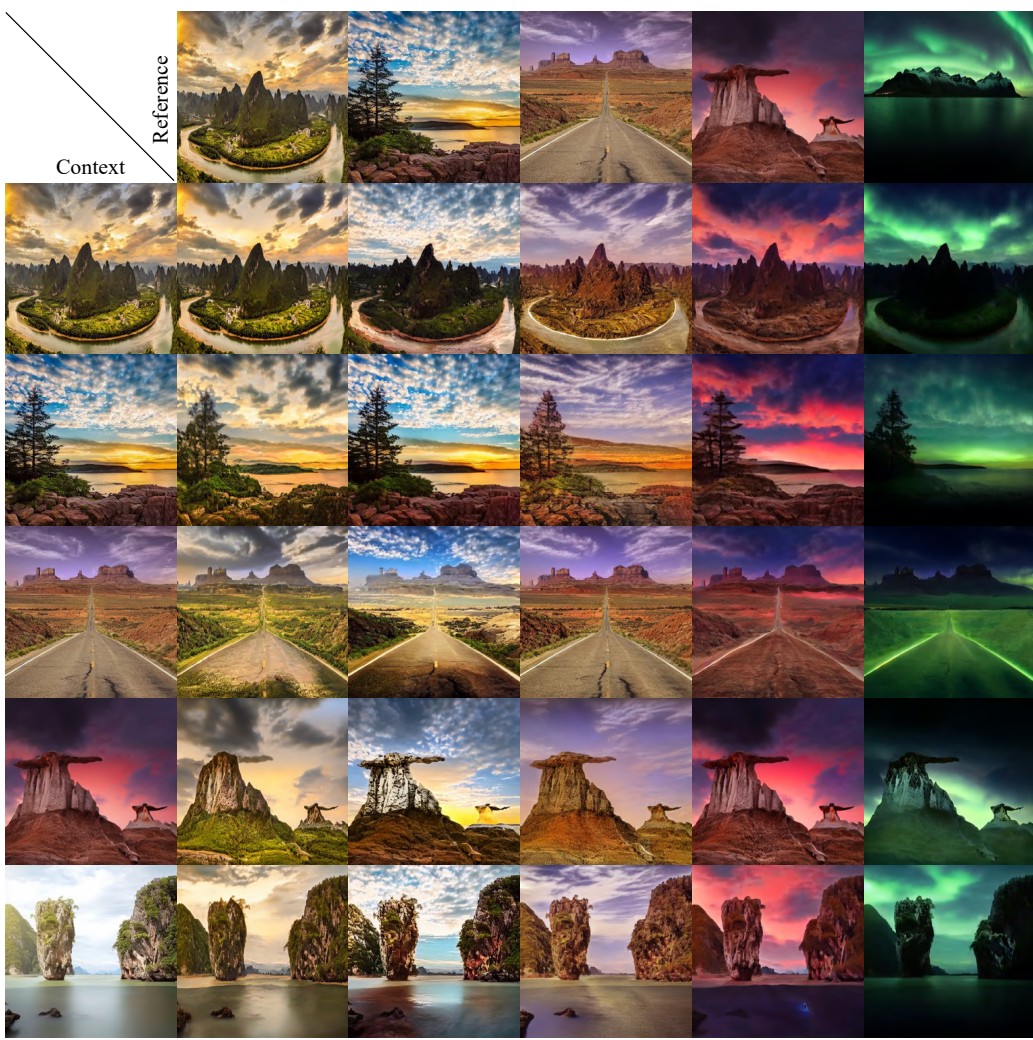

Figure 18: Additional image semantic style transfer (appearance) results on given context and reference image pairs.

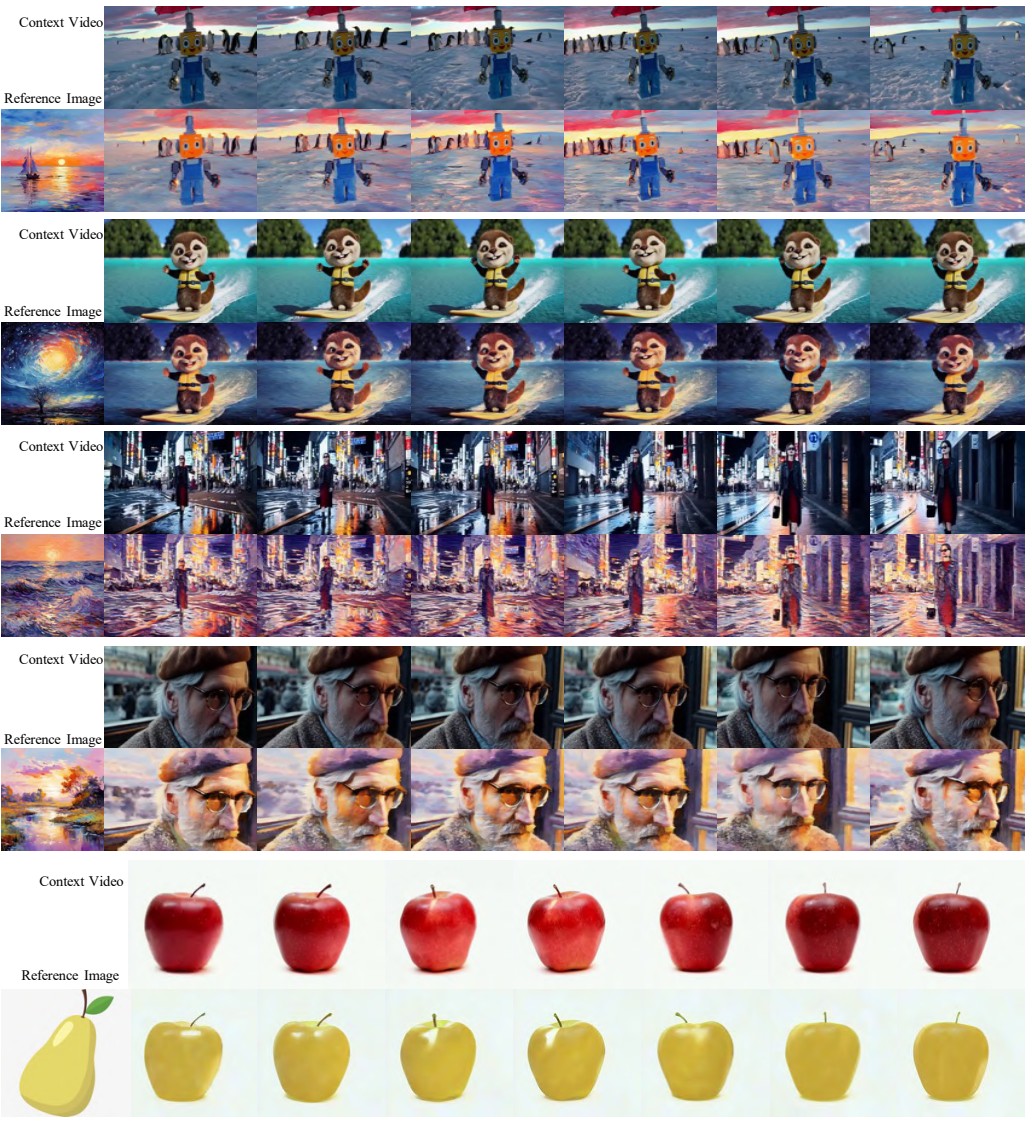

Figure 19: Additional video semantic style transfer results on given context videos and reference images.

