# OpenReview forum: "Semantix: An Energy-guided Sampler for Semantic Style Transfer"
_ICLR.cc/2025/Conference — ICLR 2025 Poster_

### Official Review · Reviewer_uDZ6 · 2024-10-28

**Soundness:** 3
**Presentation:** 3
**Contribution:** 2
**Rating:** 6
**Confidence:** 4

**Summary:**

This paper proposes a semantic style transfer method called Semantix that can handle style transfer and appearance transfer simultaneously. The key idea of this paper is to use the gradients of well-designed energy functions to guide the sampling process of pretrained diffusion models. Specifically, the three terms Style Feature Guidance, Spatial Feature Guidance, and Semantic Distance collectively form the energy function of this paper.

**Strengths:**

1. The proposed method is capable of performing both style transfer and appearance transfer with a single framework.
2. The proposed method is training-free and can be extended to video style transfer.
3. The stylized images generated by the proposed method are plausible.
4. Extensive experiments are conducted to evaluate the performance of the proposed method.

**Weaknesses:**

1. My primary concern is the lack of novelty in this paper. The use of energy function gradients to guide the diffusion sampling process has been extensively studied in prior diffusion-based research. Additionally, the energy function itself is not novel: the style feature guidance term resembles the style loss in style transfer, and the spatial feature guidance term aligns with content loss in style transfer.

2. The paper mentions style transfer for videos. What specific adaptations are made in the proposed method for video style transfer? Are motion aspects considered in both the context and synthetic images? Additionally, does applying this method to pre-trained video models impact stability and consistency? If so, why?

3. In the first row of Figure 7, the stylized images generated under different settings appear very similar, without showing significant differences.

4. In Figure 3, why is it claimed that the best results occur at t=601? The results across time steps seem quite similar. It is recommended to compare results across different time steps in the experiments.

5. I’m curious about the runtime of the proposed method. Is its inference speed comparable to or faster than previous methods?

------------------------------------------------------------------------------------

After reviewing them, most of my concerns have been addressed. Although the novelty of the proposed method may not fully meet the high expectations for ICLR, this is a strong application with impressive results. As a result, I’ve raised my score from 5 to 6, placing it in the borderline range.

**Questions:**

please see weaknesses.

---

> ### Author Response · Authors · 2024-11-18
>
> We thank for your review and helpful suggestions. Below are our clarifications of your concerns:
>
> 1. **Limited Novelty**:
>
>     a. *Energy Guided is widely used.* While energy functions have been used to guide sampling, their design remains an area for further exploration. Unlike previous methods of energy functions for various domain, we focus on refining energy function design for zero-shot semantic style transfer, which is the first work in this field to the best of our knowledge. We have thoroughly addressed the existing energy-guided approaches in the Related Work section of the paper, **e.g., Lines 181-188**.
>
>     b. *Our Energy Function design is not novel.* We consider that the energy function we proposed demonstrates significant innovation. We innovatively apply energy functions to semantic style transfer, incorporating DIFT, positional encodings (PE), and various feature distance. To the best of our knowledge, this is the first use of these methods in style or appearance transfer. Additionally, our use of L2 regularization with feature distance as a balancing term for style and appearance introduces a novel approach.
>
> 2. **Motion Consistency**:
>
>     a. *Specific adaptations for video.*  We would like to address and clarify some potential misunderstandings regarding our work. Our method is **a sampler for diffusion models**. We do not have any specific adaptations for video task. Moreover, this property is also one of the advantages of our method.
>
>     b. *Motion consistency for images.* We do not consider motion aspects **in the field of images**. In the video domain, the content video is provided by users and we do not consider the consistency or other attributes of provided video. Regarding to the synthetic video, after style transfer through a pre-trained video model, further contains the understanding of motion consistency.
>
>     c. *Apply to pretrained video model.* We wish to emphasize to the reviewer that our video experiments were conducted using a pre-trained video model. Furthermore, our proposed approach is a sampler that does not modify the pre-trained model. As a result, the motion consistency of the generated video model remains fully aligned with the motion consistency capabilities of the T2V model used, without introducing any additional damage.
>
> 3. **No significant differences in Fig. 7:** We observe that **removing any of the guiding components leads to a noticeable decline** in generation quality, as shown in the first row of Fig. 7:
>
>     a. Without style feature guidance, the style similarity between generated image and style image slightly decreases.
>
>     b. Without positional encoding, the image becomes blurry.
>
>     c. Without spatial feature guidance, the image distorts and exhibits structural degradation.
>
>     d. Without the semantic distance regularization, the generated image may overfit to style or structure, leading to visual disharmony and a reduction in image quality.
>
> 4. **Selection of timesteps**: During the experiment, we observed that at T = 601, there was an optimal trade-off between generation quality and sampling speed.
>
>     a.  T > 601, both previous studies [1-3] and Fig. 3 indicate that the image structure is formed early in the sampling process. Therefore, when T > 601, the spatial structure of the image has not fully developed, and the semantic correspondence is ambiguous. At this stage, feature guidance can disrupt the image structure, thereby reducing the quality of the generated image.
>
>     b. T < 601, the correspondence tends to be more accurate. However, too few sampling steps weaken the energy function's guidance, reducing style transfer effectiveness. Fewer timesteps with more sampling steps may also lead to overfitting due to minimal feature variation. Furthermore, as mentioned in the Appendix E, our method is robust and does not require highly precise correspondence.
>
>     c. We conducted an ablation study on the sampling timesteps, with the results shown in the Fig. 8 in the Appendix E. It also proves that the best result is achieved when T=601.
>
> 5. **Runtime of Semantix**: In **Appendix E and Table 5 of our initial version, we had already provided a detailed comparison and analysis** of the computational speed and memory consumption of our approach against those of other methods. Since our sampler is guided by energy gradients, there is a slight increase in sampling time.
>
>
> [1] Voynov, Andrey, et al. "p+: Extended textual conditioning in text-to-image generation." *arXiv preprint arXiv:2303.09522* (2023).
>
> [2] Agarwal, Aishwarya, et al. "A-star: Test-time attention segregation and retention for text-to-image synthesis." *Proceedings of the IEEE/CVF International Conference on Computer Vision*. 2023.
>
> [3] Agarwal, Aishwarya, et al. "An image is worth multiple words: Multi-attribute inversion for constrained text-to-image synthesis." *arXiv preprint arXiv:2311.11919* (2023).

---

> ### Author Response · Authors · 2024-11-25
> **A Kind Request for Feedback**
>
> Dear Reviewer uDZ6,
>
> Thank you once again for your valuable feedback and comments! We have addressed each of your concerns individually. Could you kindly confirm whether they have resolved your issues?  With the deadline for the discussion approaching, we would greatly appreciate it if you could let us know whether our response meets your expectations and update our overall rating accordingly.
>
>
> Sincerely,
>
> Semantix team

---

> > ### Comment · Reviewer_uDZ6 · 2024-11-27
> > **Feedback after reading the responses.**
> >
> > Thank you for your point-to-point responses. After reviewing them, most of my concerns have been addressed. Although the novelty of the proposed method may not fully meet the high expectations for ICLR, this is a strong application with impressive results. As a result, I’ve raised my score from 5 to 6, placing it in the borderline range.

---

### Official Review · Reviewer_ashT · 2024-11-01

**Soundness:** 3
**Presentation:** 3
**Contribution:** 2
**Rating:** 5
**Confidence:** 4

**Summary:**

This paper introduces an approach to transferring style between a reference image/video and a context image/video while accounting for the semantics of the elements observed in the images. This is achieved by providing guidance in a diffusion model in the form of an energy function. The proposed energy function leverages semantic correspondences between the reference and context inputs while preserving spatial relationships.

**Strengths:**

- Using semantic correspondence to perform style transfer is intuitive
- The training-free aspect of the method makes it attractive
- The results are visually convincing
- The paper is generally clear and reasonably easy to follow (with some exceptions listed in Weaknesses)

**Weaknesses:**

1) Originality:
- The idea of accounting for semantics in style transfer has been studied in the past. The most closely related work is Ozaydin et al. TMLR 2024, DSI2I: Dense Style for Unpaired Exemplar-based Image-to-Image Translation, which also uses the notion of semantic correspondence. However, other references therein already looked into leveraging semantics for style transfer (Shen et al., CVPR 2019; Bhattacharjee et al., CVPR 2020; Jeong et al, CVPR 2021; Kim et al. CVPR 2022). I acknowledge that there are methodological differences between these works and the proposed method, but this limits the conceptual originality of this submission, and this line of research should be discussed.

2) Methodology:
- The choice of T=601 seems a bit arbitrary. Does this work for any scenario/generative model? Is the influence of this studied empirically (this does not seem to be part of the ablation study)?
- The method has a number of hyper-parameters (\omega in Eq. 4, the \gammas in Eq. 5, \lambda in Eqs. 9-10), which seem to be set to fairly precise values according to Appendix B. Their influence is to some degree studied (in a on-off manner in Table 4, and on one particular example in Fig 12), but this analysis could be made more thorough.
- Line 313-314: This statement is not clear. How are the object region masks used?
- Won’t the spatial guidance of Eq. 12 tend to prevent the style transfer, i.e., by encouraging to preserve the appearance from the context image?
- It seems that the semantic aspect of the transfer is achieved via both the semantic correspondences used in F_{ref}, and the semantic distance employed in F_{reg}. Why is either one of these not sufficient, i.e., why are both needed?
- In Eq. 14, does sg stand for gradient clipping (as argued below the equation), or stop gradient (as would seem more intuitive based on the name)?
- As I understand it, in the case of video, motion consistency is assumed to be inherent to the generative model itself. However, even if it is the case, is there any guarantee that the style transfer process won’t destroy, or at least damage, this motion consistency?

Experiments:
- From Table 1, the proposed method offers a good tradeoff between the different metrics. However, so do StyleID and, to some degree, StyleFormer. Nevertheless, I acknowledge that the qualitative results show that the proposed method yields better quality than these baselines. This raises the question of whether these qualitative examples were cherry-picked to highlight the better performance of the proposed method over StyleID and StyleFormer, or if the metrics are far from being perfect. I acknowledge that the user study of Table 3 helps clarifying things.

Clarity:
- I find the overview of the method, both in the introduction and in Section 4, hard to follow.
- The purpose of the preliminaries (Section 3) is not clear, with mathematical notation sometimes not explicitly defined (e.g., x_t, \phi, \theta), and the link to the proposed method missing.

Minor:
- The authors should use \citep instead of \cite, unless the names of the authors of the corresponding reference should be part of the sentence.

**Questions:**

Please see Weaknesses.

---

> ### Author Response · Authors · 2024-11-17
>
> We sincerely appreciate the insightful comments from *Reviewer ashT*, which will surely enhance the quality of our paper. In response to your concerns, we would like to provide the following clarifications:
>
> 1. **Concept Novelty**: We have included additional discussion in our revised version **Lines 164-173**. Overall, while the related work [1] mentioned by the reviewer also introduces the concept of semantic style transfer, our definitions of semantics and the task of semantic style transfer differ significantly. We would like to elaborate on our innovations from the following perspectives:
>
>     a. In terms of semantics, the key distinction lies in the extraction and utilization of semantic features. Unlike prior works that rely on segmentation labels [2, 3] or DSI2I [1], which leverages CLIP visual branch for dense semantic feature reuse but loses alignment with the natural language dimension, our method extracts semantic features with a natural language dimension from the pretrained T2I or T2V models. These features are then employed within an energy function to effectively guide semantic style transfer.
>
>     b. In terms of tasks, DSI2I [1] is an I2I task that only extracts dense semantics and reuse it in given image. Instead, our work first defines a more expansive zero-shot semantic style transfer task, which is focused on generation. Rather than merely reusing the low-level features of a given image, e.g., color, we leveraged a powerful pre-trained diffusion model to generate novel style images that adhere to semantic constraints derived from the features of the given image.
>
>     c. To the best of our knowledge, we are the first to unify style and appearance transfer through semantic understanding and mapping in visual generation tasks. Our work also innovatively introduces energy-guided semantic alignment for style transfer.
>
> 2. **Selection of timesteps**: during the experiment, we observed that at T = 601, there was an optimal trade-off between generation quality and sampling speed.
>
>     a.  T > 601, both previous studies[4-6] and Fig. 3 indicate that the image structure is formed early in the sampling process. Therefore, when T > 601, the spatial structure of the image has not fully developed, and the semantic correspondence is ambiguous. At this stage, feature guidance can disrupt the image structure, thereby reducing the quality of the generated image.
>
>     b. T < 601, the correspondence tends to be more accurate. However, too few sampling steps weaken the energy function's guidance, reducing style transfer effectiveness. Fewer timesteps with more sampling steps may also lead to overfitting due to minimal feature variation. Furthermore, as mentioned in the Appendix E, our method is robust and does not require highly precise correspondence.
>
>     c. We conducted an ablation study on the sampling timesteps, with the results shown in the Fig. 8 in the Appendix E. It also proves that the best result is achieved when T=601.
>
> 3. **Selection of Hyper-parameters**: $\omega$ is actually set as the widely used CFG value. And hyper-parameters in Eq. 5 are set heuristically based on proportion. The energy guidance functions are also similarly to CFG. Ablation studies in Fig. 13 show that performance is not highly sensitive to these hyper-parameters($\gamma$, $\lambda$).
>
>
> [1] Ozaydin, Baran, et al. "DSI2I: Dense Style for Unpaired Exemplar-based Image-to-Image Translation." *Transactions on Machine Learning Research*.
>
> [2] Shen, Zhiqiang, et al. "Towards instance-level image-to-image translation." *Proceedings of the IEEE/CVF conference on computer vision and pattern recognition*. 2019.
>
> [3] Bhattacharjee, Deblina, et al. "Dunit: Detection-based unsupervised image-to-image translation." *Proceedings of the IEEE/CVF Conference on Computer Vision and Pattern Recognition*. 2020.
>
> [4] Voynov, Andrey, et al. "p+: Extended textual conditioning in text-to-image generation." arXiv preprint arXiv:2303.09522 (2023).
>
> [5] Agarwal, Aishwarya, et al. "A-star: Test-time attention segregation and retention for text-to-image synthesis." Proceedings of the IEEE/CVF International Conference on Computer Vision. 2023.
>
> [6] Agarwal, Aishwarya, et al. "An image is worth multiple words: Multi-attribute inversion for constrained text-to-image synthesis." arXiv preprint arXiv:2311.11919 (2023).

---

> ### Author Response · Authors · 2024-11-17
>
> 4. **Clarification regarding Object Region**: In our revised manuscript, we have included more detailed explanations of the procedure for obtaining the object region mask, as highlighted content in **Lines 366–368**. Specifically,
>
>     a. Our masks are obtained as described in [1]. In brief, during the inversion process, we apply $k$-means clustering on the self-attention maps to generate the segment masks. For further details, please refer to the original paper [1].
>     b. Apply: After obtaining the masks, we apply them to the features, performing calculations and guidance only within the masked regions.
>
> 5. **Conflict of Different Guidance**: Spatial guidance and style guidance interact with each other, and during the sampling process, the results may favor the spatial reference over the style (or vice versa). To address this issue, we introduce semantic distance as a mitigating mechanism. Ablation experiments show that the proposed semantic distance term effectively avoids overfitting and balances style and spatial feature guidance. Therefore, incorporating semantic distance helps prevent spatial feature guidance from negatively impacting style transfer (or vice versa). For the detail of how Semantic Distance works, please refer to our response to other reviewers comments, and the added sections in our revised manuscript, e.g., **Line 385-388**.
>
> 6. **Necessary of both terms**: We would like to clarify that the semantic aspect of the transfer is actually achieved through style guidance. The regularization term serves to balance the interplay between style and spatial guidance. More specifically, when the style and context images exhibit weak semantic correspondence, the semantic guidance in the regularization term can further promote the injection of style features. Furthermore, when the generated image overfits the context image, the regularization term enhances style injection through semantic strengthening, achieving a balance between style and context features. Overall, both terms are indispensable for achieving a balanced semantic style transfer.
>
> 7. **Typo of explanation of sg($\cdot$)**: Thanks for pointing that out. "sg" here refers to stop gradient. We have revised it in manuscript.
>
> 8. **Guarantee of video motion consistency**: Compared to other frame-based video editing frameworks, our model, which is based on a pre-trained T2V model, ensures stronger consistency. Furthermore, our proposed approach is a sampler that does not modify the pre-trained model. As a result, the motion consistency of the generated video model remains fully aligned with the motion consistency capabilities of the T2V model used, without introducing any additional damage.
>
> 9. **Clarify the demonstrated results**: We are committed to transparency and accuracy in our research and do not use cherry-picking to exaggerate our model's capabilities. All qualitative experiments were conducted without selection bias, and for the quantitative experiments, we used the same benchmark datasets as studies like StyleID [2], where cherry-picking is not feasible. Our code is available in the supplementary file, and we encourage reviewers to verify the results. Furthermore, we plan to release the code publicly once the paper is published.
>
> 10. **Clear presentation**: For some ambiguous descriptions in the manuscript, we have rephrased them in the revised version, which can be found in the highlighted sections of the newly submitted manuscript.
>
> 11. **Symbol notation**: Thanks for the suggestion. We have added the definitions of mathematical notation in the revised manuscript.
>
> 12. **\cite format**: Thanks for raising this issue. We have made the appropriate corrections in the revised manuscript.
>
> **Finally, we sincerely thank the reviewer once again for their valuable feedback, which has greatly enhanced the clarity and presentation of our work. If you have any further questions regarding our responses, we kindly invite you to raise them during the discussion phase.**
>
> [1] Patashnik, Or, et al. "Localizing object-level shape variations with text-to-image diffusion models." Proceedings of the IEEE/CVF International Conference on Computer Vision. 2023.
>
> [2] Chung, Jiwoo, Sangeek Hyun, and Jae-Pil Heo. "Style injection in diffusion: A training-free approach for adapting large-scale diffusion models for style transfer." Proceedings of the IEEE/CVF Conference on Computer Vision and Pattern Recognition. 2024.

---

> ### Author Response · Authors · 2024-11-25
> **A Kind Request for Feedback**
>
> Dear Reviewer ashT,
>
> Thank you once again for your valuable feedback and comments! With the deadline for the discussion approaching, we would greatly appreciate it if you could let us know whether our response meets your expectations and update our overall rating accordingly. Please don’t hesitate to let us know if there are any remaining concerns, we would be glad to address them.
>
> Sincerely,
>
> Semantix team

---

> > ### Comment · Reviewer_ashT · 2024-11-26
> > **Feedback**
> >
> > Thank you for the detailed answers to my comments. While these answers addressed my more technical comments, I remain on the borderline with this work. Altogether, I acknowledge that it represents a nice engineering work, with convincing results, but I would expect some more scientific novelty/insights in an ICLR paper.

---

### Official Review · Reviewer_84Mq · 2024-11-05

**Soundness:** 3
**Presentation:** 3
**Contribution:** 3
**Rating:** 8
**Confidence:** 5

**Summary:**

The paper introduces Semantix, an energy-guided sampler designed for semantic style transfer that aims to transfer style and appearance features from a reference image to a target visual content based on semantic correspondence. The method leverages pre-trained diffusion models' semantic understanding capabilities to guide both style and appearance transfer. The approach consists of three key components: Style Feature Guidance for aligning style features, Spatial Feature Guidance for maintaining spatial coherence, and Semantic Distance as a regularization term. After inverting reference and context images/videos to noise space using SDEs, Semantix utilizes these components within a carefully crafted energy function to guide the sampling process.

**Strengths:**

1. The method is easy and can be easily applied to both image and video models without requiring additional modifications.

2. The experimental results show that Semantix outperforms existing solutions in both style and appearance transfer tasks while maintaining structural integrity and semantic consistency.


3. The writing is mostly clear and the visualizations are decent. For example, I especially like the analysis presented in Fig. 3.

**Weaknesses:**

1. The motivation for semantic distance is not intuitive. The authors claim that this regularization is meant to prevent overfitting. It is unclear what overfitting the authors indicate, such as to prevent overfitting to the context or reference, or both? Also, it is hard to reason about why the regularization works to handle the problem.

2. Important reference is missing. For example, [1] also deals with universal style transfer which applies to both image and video domains, which is an ideal comparison and discussion.

3. Positional encoding doesn’t seem to have much impact. The qualitative difference between the default version and the deprived version is small. The quantitative difference is also small.

4. Demonstration problems: 1) Fig. 2 is very uninformative. The rightmost part is hard to perceptually understand. The leftmost part does not convey much information. The middle part is hard to observe different arrows and get their exact meanings. 2) Some tables have varied font sizes for titles, such as Tab. 2, 3, 4.

[1] Wu, Zijie, et al. "Ccpl: Contrastive coherence preserving loss for versatile style transfer." European Conference on Computer Vision. Cham: Springer Nature Switzerland, 2022.

**Questions:**

Why do you revert the images at step 601 rather than smaller steps since from my point of view, smaller steps seem to induce better correspondence?

---

> ### Author Response · Authors · 2024-11-17
>
> We sincerely appreciate the strong recognition and encouragement from *Reviewer 84Mq*, which validates our dedication to contributing meaningfully to the community. We have carefully reviewed your comments and incorporated revisions to address them and enhance the clarity of our work.
>
> 1. **Motivation of Semantic Distances**: Semantic distance is introduced to balance style and context features, preventing structural distortion and style deviation.  As illustrated in Figure 7, the manifestations of overfitting can be observed. Without semantic distance, the images in the first row (boy and oil painting) excessively align with the style image (the facial features and clothing of the generated boy are as blurred as those of the woman in the oil painting), while those in the second row (black suit jacket and red pants) overfit to the context image (the suit jacket color become dark instead of red). Further unbalance results can be found in appendix.
>
>     Technically, our semantic distance term serves as an L2 regularization. During the sampling process, we aim to optimize the proposed energy function. The style guidance and spatial guidance are treated as key components of the energy function, while the semantic distance serves as a regularization term. It is worth noting that the parameters of Style guidance or Spatial guidance are actually the corresponding feature vectors, while the Swapped Cross Attention feature contains the output of style guidance and spatial guidance.  By constraining and penalizing the feature vectors involved in the computation of the regular terms, e.g. Style guidance and Spatial guidance, it ensures that the style and spatial feature vectors remain within a more compact space. This leads to a smoother solution space, reduces the risk of overfitting, and ensures stability throughout the sampling process.
>
> 2. **Missing Reference**: Thank you for your valuable reminder. We have adopted this work [1] as a baseline and conducted comprehensive qualitative and quantitative comparisons with it on both image and video tasks. The quantitative results are provided in the Tables below:
>
> Table 1: Quantitative Comparison of Video Style Transfer
>
> | Method    | LPIPS ↓ | CFSD ↓ | SSIM ↑ | Gram Metrics $_{\times 10^2}$ ↓ | PickScore ↑ | HPS ↑  |
> |:-----------:|:----------:|:---------:|:---------:|:----------------------:|:-------------:|:--------:|
> | **CCPL[1]**  | 0.523    | 0.133   | 0.536   | 4.861                | 16.75       | 16.79  |
> | **Ours**      | **0.461**    | **0.117**   | **0.589**   | **2.524**                | **19.95**       | **18.78**  |
>
> Table 2: Quantitative Comparison of Video Style Transfer
>
> | Method | Semantic Consistency ↑ | Object Consistency ↑ | Motion Alignment ↑ | Visual Quality ↑ | Motion Quality ↑ | Temporal Consistency ↑ |
> |:-----------:|:-------:|:---------:|:---------------------:|:-------------------:|:-------------------:|:--------------------------:|
> | **CCPL[1]** | 0.942                   | 0.943                 | **-1.792**              | 48.92             | 53.25             | 59.64                    |
> | **Ours**      | **0.944**                   | **0.955**                 | -1.894              | **55.86**             | **53.99**             | **60.05**                    |
>
> For qualitative results and more quantitative comparisons, please refer to the revised manuscript, **e.g. Figs. 4,5 & Tables 1,2**. The experimental results show that our method outperforms [1] in both image and video tasks.
>
> 3. **Impact of Position Encoding**: In our proposed method, positional encoding plays a crucial role for the following reasons:
>
>     a. Position encoding encodes relative spatial information, which helps maintain structural integrity. Without it, generated images may be blurry or lose the relative position of features as shown in the Fig.7.
>
>     b. In cases where semantic correspondence is ambiguous, we can control the strength of the positional encoding to achieve accurate spatial alignment, thereby preventing a degradation in generation quality.
>
> [1] Wu, Zijie, et al. "Ccpl: Contrastive coherence preserving loss for versatile style transfer." European Conference on Computer Vision. Cham: Springer Nature Switzerland, 2022.

---

> ### Author Response · Authors · 2024-11-17
>
> 4. **Demonstration problems**: We have reformatted Tab.2,3,4 and addressed the issue of varied font sizes. We will further optimize the Fig.2  in the camera-ready version to enhance its clarity.
>
> 5. **Selections of inversion timesteps**: During the experiment, we observed that at T = 601, there was an optimal trade-off between generation quality and sampling speed.
>
>     a. T > 601, both previous studies[1-3] and Fig.3 indicate that the image structure is formed early in the sampling process. Therefore, when T > 601, the spatial structure of the image has not fully developed, and the semantic correspondence is ambiguous. At this stage, feature guidance can disrupt the image structure, thereby reducing the quality of the generated image.
>
>     b. T < 601, the correspondence tends to be more accurate. However, too few sampling steps weaken the energy function's guidance, reducing style transfer effectiveness. Fewer timesteps with more sampling steps may also lead to overfitting due to minimal feature variation. Furthermore, as mentioned in the Appendix E, our method is robust and does not require highly precise correspondence.
>
>     c. We conducted an ablation study on the sampling timesteps, with the results shown in the Fig. 8 in the Appendix E. It also proves that the best result is achieved when T=601.
>
> **We sincerely appreciate your kind and encouraging feedback again. If you have any further questions about our work or responses, we kindly invite you to raise them during the discussion phase.**
>
> [1] Voynov, Andrey, et al. "p+: Extended textual conditioning in text-to-image generation." *arXiv preprint arXiv:2303.09522* (2023).
>
> [2] Agarwal, Aishwarya, et al. "A-star: Test-time attention segregation and retention for text-to-image synthesis." *Proceedings of the IEEE/CVF International Conference on Computer Vision*. 2023.
>
> [3] Agarwal, Aishwarya, et al. "An image is worth multiple words: Multi-attribute inversion for constrained text-to-image synthesis." *arXiv preprint arXiv:2311.11919* (2023).

---

> > ### Comment · Reviewer_84Mq · 2024-11-27
> > **Feedback after reading authors' responses**
> >
> > I sincerely thank the authors for the efforts of posting these responses. My concerns are addressed, including the comparison to CCPL. I don't have further comments right away. So I will maintain my score and actively check other reviewers' comments.

---

### Official Review · Reviewer_thdC · 2024-11-05

**Soundness:** 3
**Presentation:** 2
**Contribution:** 3
**Rating:** 6
**Confidence:** 4

**Summary:**

This paper proposes a training-free semantic style transfer method that innovatively leverages an energy-guided sampler, enabling it to produce both image and video results that surpass existing methods. The task of semantic style transfer is introduced to emphasize the alignment of semantic relationships,  enabling style and appearance transfer while avoiding content degradation and structural disruption.

**Strengths:**

1. Style Feature Guidance, Spatial Feature Guidance and Semantic Distance are clearly defined and correspond to the theme, successfully achieving semantic style transfer.
2. The paper is well-written and shows the model’s effectiveness through both quantitative evaluations and user studies.
3. The authors provide extensive supplementary materials and appendices to validate the effectiveness of the proposed method.

**Weaknesses:**

1. Some expression in Methods section lacks logical coherence. Some innovations are introduced rather abruptly, with insufficient theoretical foundation and clear explanations, such as the motivation of spatial feature guidance.
2. The related work section focuses on the field of style transfer but lacks a discussion of energy functions and diffusion-based inference guidance.

**Questions:**

1. I have some confusion regarding spatial feature guidance. Why does calculating the similarity between output features and content features in the feature map preserve spatial structure? Would this impact the effectiveness of style guidance, such as injecting style information from the content reference?
2. Regarding semantic distance, it seems that you have borrowed from previous methods ("Visual Style Prompting with Swapping Self-Attention") in the self-attention part. However, why does directly computing the L2 distance in the cross-attention mechanism help prevent overfitting? It would be helpful to explain the rationale behind this approach.
3. Acquiring feature maps requires three inference processes in total. Would this significantly increase inference time?

---

> ### Author Response · Authors · 2024-11-17
>
> We sincerely appreciate the recognition and the valuable comments from *Reviewer thdC*. Your feedback has greatly contributed to improving the quality of our submission. We kindly request you to review our responses to your suggestions:
>
> 1. **Lacks logical coherence**: we would provide a summary of the overall motivation and purpose of our method, as well as to clarify its underlying logic.
>
>     Given a pre-trained diffusion model and a set of noise signals from DDPM inversion of the input images, we propose Semantix, an energy-guided sampler, to achieve zero-shot training-free semantic style transfer. The energy-guided sampler is defined by three distinct guidance components, as outlined below:
>
>     a. Style Feature Guidance: To inject style features into the context image, we introduce style feature guidance, which minimizes the feature distance between the output and style features for effective style transfer.
>
>     b. Spatial Feature Guidance: To avoid disrupting the spatial structure of the generated image during style feature injection, we introduce spatial feature guidance, which preserves the spatial integrity of generated image.
>
>     c. Semantic Distance: To prevent overfitting to style or context features and to maintain an effective balance between style and structure, we introduce a semantic distance regularization term, which enhances the overall image quality.
>
>     These three components work synergistically. Style feature guidance addresses the core task of semantic style transfer, which is the alignment of semantic styles, while spatial feature guidance ensures structural consistency between the generated image and the given image. Finally, semantic distance functions as a regularization term, mitigating overfitting to either style or spatial features.
>
> 2. **Insufficient theoretical foundation and clear explanations**: we explain the spatial feature guidance in detail below
>
>     Spatial feature guidance aims to preserve the consistency of the image spatial structure before and after style transfer.
>
>     Previous studies [1, 2] have shown that features in diffusion models contain spatial information and some methods [3,4] achieve context preservation by replacing these features. However, such approaches lack a mechanism to ensure consistency when utilizing these features, which can easily cause a mismatch in distribution between the introduced and original features, resulting in artifacts or highlight distortions in the generated images.
>
>     In contrast, our approach preserves spatial structure by minimizing the distance between context and output features. Specifically, we compute the feature distance at corresponding positions during sampling and design an energy function to guide the sampling, aligning spatial structures and maintaining spatial integrity.
>
>     We have also updated our manuscript to clarify and provide more specific descriptions of the content related to other methods. Kindly review the **red** part in the updated draft, e.g., **Lines 370–377**. The expression will be further improved in the camera-ready version.
>
> [1] Patashnik, Or, et al. "Localizing object-level shape variations with text-to-image diffusion models." Proceedings of the IEEE/CVF International Conference on Computer Vision. 2023.
>
> [2] Tumanyan, Narek, et al. "Plug-and-play diffusion features for text-driven image-to-image translation." Proceedings of the IEEE/CVF Conference on Computer Vision and Pattern Recognition. 2023.
>
> [3] Chung, Jiwoo, Sangeek Hyun, and Jae-Pil Heo. "Style injection in diffusion: A training-free approach for adapting large-scale diffusion models for style transfer." Proceedings of the IEEE/CVF Conference on Computer Vision and Pattern Recognition. 2024.
>
> [4] Alaluf, Yuval, et al. "Cross-image attention for zero-shot appearance transfer." ACM SIGGRAPH 2024 Conference Papers. 2024.

---

> ### Author Response · Authors · 2024-11-17
>
> 3. **Lack of Literatures**: In the revised version, we provide a literature review on the energy function, including the following keys:
>
>     a. Theory of energy functions. Previous research interprets diffusion models as energy-based models [2], where the energy function guides and controls the generation process for precise visual outputs.
>
>     b. Applications of energy functions. The energy function has a wide range of applications. Some methods use the energy function to guide image editing [3-5] and translation [6]. The energy function also shows significant potential in controllable generation, guiding generation through conditions such as sketch [7], mask [8], layout [9], concept [2], and universal guidance [10,1], enabling precise control over the output.
>
>     The corresponding content has also been updated in the draft. We kindly invite you to review the highlighted parts in the “Related Works” section, **Lines 181–188**.
>
> 4. **Clarification regarding spatial feature guidance**:
>
>     a. *Why does it work?* Previous studies [11, 12] have shown that features within diffusion models contain spatial information, and some methods [13, 14] use feature replacement for visual transfer and can preserve structure. In contrast, our approach minimizes the distance between context and output features to guide the generation process, preserving the original spatial structure. Specifically, we calculate the feature distance at corresponding spatial locations during sampling and use an energy function to align the spatial structure features, ensuring effective preservation of spatial integrity.
>
>     b. *Would it impact the other guidance?* The answer is yes. To prevent the impact to the other guidance after introducing spatial and style feature guidance, we proposed a **semantic distance regularization term**. Ablation studies demonstrate that semantic distance term effectively mitigates the negative impact and maintains a balance between style and spatial feature guidance.
>
> [1] Yu, Jiwen, et al. "Freedom: Training-free energy-guided conditional diffusion model." *Proceedings of the IEEE/CVF International Conference on Computer Vision*. 2023.
>
> [2] Liu, Nan, et al. "Compositional visual generation with composable diffusion models." *European Conference on Computer Vision*. Cham: Springer Nature Switzerland, 2022.
>
> [3] Mou, Chong, et al. "Dragondiffusion: Enabling drag-style manipulation on diffusion models." *arXiv preprint arXiv:2307.02421* (2023).
>
> [4] Mou, Chong, et al. "Diffeditor: Boosting accuracy and flexibility on diffusion-based image editing." *Proceedings of the IEEE/CVF Conference on Computer Vision and Pattern Recognition*. 2024.
>
> [5] Epstein, Dave, et al. "Diffusion self-guidance for controllable image generation." *Advances in Neural Information Processing Systems* 36 (2023): 16222-16239.
>
> [6] Zhao, Min, et al. "Egsde: Unpaired image-to-image translation via energy-guided stochastic differential equations." *Advances in Neural Information Processing Systems* 35 (2022): 3609-3623.
>
> [7] Voynov, Andrey, Kfir Aberman, and Daniel Cohen-Or. "Sketch-guided text-to-image diffusion models." *ACM SIGGRAPH 2023 Conference Proceedings*. 2023.
>
> [8] Singh, Jaskirat, Stephen Gould, and Liang Zheng. "High-fidelity guided image synthesis with latent diffusion models." *2023 IEEE/CVF Conference on Computer Vision and Pattern Recognition (CVPR)*. IEEE, 2023.
>
> [9] Chen, Minghao, Iro Laina, and Andrea Vedaldi. "Training-free layout control with cross-attention guidance." *Proceedings of the IEEE/CVF Winter Conference on Applications of Computer Vision*. 2024.
>
> [10] Bansal, Arpit, et al. "Universal guidance for diffusion models." *Proceedings of the IEEE/CVF Conference on Computer Vision and Pattern Recognition*. 2023.
>
> [11] Patashnik, Or, et al. "Localizing object-level shape variations with text-to-image diffusion models." Proceedings of the IEEE/CVF International Conference on Computer Vision. 2023.
>
> [12] Tumanyan, Narek, et al. "Plug-and-play diffusion features for text-driven image-to-image translation." Proceedings of the IEEE/CVF Conference on Computer Vision and Pattern Recognition. 2023.
>
> [13] Chung, Jiwoo, Sangeek Hyun, and Jae-Pil Heo. "Style injection in diffusion: A training-free approach for adapting large-scale diffusion models for style transfer." Proceedings of the IEEE/CVF Conference on Computer Vision and Pattern Recognition. 2024.
>
> [14] Alaluf, Yuval, et al. "Cross-image attention for zero-shot appearance transfer." ACM SIGGRAPH 2024 Conference Papers. 2024.

---

> ### Author Response · Authors · 2024-11-17
>
> 5. **Clarification regarding semantic distance**:
>
>     a. We firstly discussed the mentioned literature [1] in our manuscript. While both our method and theirs employ swapping self-attention strategy, there are differences between the two approaches in both terms of tasks and methods. In terms of tasks, their work focuses on text-based style generation, whereas our approach targets general semantic style transfer based on images, with applicability to both images and videos. In terms of computation, they inject style features by directly replacing the style features in the self-attention to achieve style transfer. In contrast, our method replaces the style features and then computes the L2 distance as an energy function to guide the generation, rather than directly injecting the features.
>
>     b. Our semantic distance term serves as an L2 regularization in this context. During the sampling process, we aim to optimize the proposed energy function. The style guidance and spatial guidance are treated as key components of the energy function, while the semantic distance serves as a regularization term. It is worth noting that the parameters of Style guidance or Spatial guidance are actually the corresponding feature vectors, while the Swapped Cross Attention feature contains the output of style guidance and spatial guidance. By constraining and penalizing the feature vectors involved in the computation of the regular terms, it ensures that the style and spatial feature vectors remain within a more compact space. This leads to a smoother solution space, reduces the risk of overfitting, and ensures stability throughout the sampling process.
>
> 6. **Inference Cost**:  It will not significantly increase the inference time, as we provide the sampling time in the Tab. 5. All feature maps are by-products of the single inference process and do not require additional inference. We can obtain all of them in the inversion process.
>
> **Finally, we sincerely appreciate your valuable suggestions and comments, which have helped us improve the clarity and completeness of our presentation. We hope our responses have adequately addressed your concerns. Should you have any further points, we kindly invite you to share them with us during the discussion phase.**
>
> [1] Jeong, Jaeseok, et al. "Visual Style Prompting with Swapping Self-Attention." arXiv preprint arXiv:2402.12974 (2024).

---

> > ### Comment · Reviewer_thdC · 2024-11-27
> > **Feedback**
> >
> > Thank you to the authors for providing a detailed response, which has addressed my concerns regarding the paper. I have no further comments and will maintain my original evaluation.

---

### Public Comment · ~Chris.C1 · 2025-09-18
**Code Consultation**

Hello authors, the code of Semantix is still unavailable now. When do you plan to release the code on github?

---

> ### Public Comment · ~Minghui_Hu1 · 2025-09-19
>
> Hi. Actually you can find runable experimental codes in attachment.

---

> > ### Public Comment · ~Chris.C1 · 2025-09-20
> >
> > Thank you for your reply! This will be of great help to me!

---

### Meta-Review · Area_Chair_c9RJ · 2024-12-17

**Metareview:**

This paper presents Semantix, a training-free method for semantic style transfer using energy function gradients with pre-trained diffusion models. It effectively integrates Style Feature Guidance, Spatial Feature Guidance, and Semantic Distance for both image and video applications, producing visually compelling results validated through extensive experiments. While the approach builds on existing concepts, its practical potential and qualitative outcomes make it a valuable contribution, with room for improvement in parameter analysis and theoretical foundation. Overall, the work shows potential, and the reviewers provided generally positive feedback. I recommend accepting this work.

**Additional Comments On Reviewer Discussion:**

During the rebuttal period, reviewers raised concerns about novelty, coherence, insufficient literature coverage, parameter selection, and motion consistency in videos. The authors addressed these by clarifying their energy function design, providing detailed literature reviews, explaining semantic distance and spatial feature guidance, and conducting additional ablation studies. They also justified timestep selection and highlighted their method’s alignment with pre-trained models to ensure video motion consistency. These responses improved clarity and addressed key concerns, demonstrating the method's robustness and practical potential. While originality remains moderate, the work's strong results and clear responses justify a positive recommendation.

---

### Decision · Program_Chairs · 2025-01-22

Accept (Poster)